# Curvature-Guided Task Synergy for Skeleton based Temporal Action Segmentation

**Guozhang Li, Xinran Duan, Mei Wang** [*]**, Lizhi Wang, Hua Huang**
School of Artificial Intelligence, Beijing Normal University, Beijing
BeijingKey Laboratory of Aritifical Intelligence for Education
Engineering Research Center of Intelligent Technology and Educational Application, Ministry of Education

## Abstract

Fine-grained temporal action segmentation plays a vital role in comprehensive human behavior understanding, with skeleton-based approaches (STAS) gaining prominence for their privacy and robustness. A core challenge in STAS arises from the conflicting feature requirements of action classification (demanding temporal invariance) and boundary localization (requiring temporal sensitivity). Existing methods typically adopt decoupled pipelines, unfortunately overlooking the inherent semantic complementarity between these sub-tasks, leading to information silos that prevent beneficial cross-task synergies. To address this challenge, we propose CurvSeg, a novel approach that synergizes classification and localization within the STAS domain through a unique geometric curvature guidance mechanism. Our key innovation lies in exploiting curvature properties of well-learned classification representations on skeleton sequences. Specifically, we observe that high curvature within action segments and low curvature at transitions effectively serve as geometric priors for precise boundary detection. CurvSeg establishes a virtuous cycle: localization predictions, guided by these curvature signals, in turn dynamically refine the classification feature space to organize into a geometry conducive to clearer boundaries. To compute stable curvature signals from potentially noisy skeleton features, we further develop a dual-expert weighting mechanism within a Mixture of Experts framework, providing task-adaptive feature extraction. Comprehensive experiments demonstrate that CurvSeg significantly enhances STAS performance across multiple benchmark datasets, achieving superior results and validating the power of geometric-guided task collaboration for this specific problem. CurvSeg

## 1 Introduction

Temporal Action Segmentation (TAS), which precisely assigns action labels to every frame in untrimmed videos, is a fundamental task for fine-grained human behavior understanding. While video-based methods (Farha & Gall, 2019; Li et al., 2023b; Yi et al., 2021; Du et al., 2022; Zhao & Song, 2022; Ren et al., 2025; Liu et al., 2023b; Li et al., 2023a) have achieved great progress, they often face challenges in scenarios demanding high privacy and robustness to varying appearances or visual noise. Skeleton-based TAS (STAS) thus emerges as a vital alternative, modeling pure motion dynamics to inherently ensure privacy and decouple from visual confounders, making it crucial for sensitive domains like healthcare.

A central challenge in STAS is the fundamental tension between its two sub-tasks: action classification and boundary localization. Classification requires temporally invariant, abstract features to ensure consistent recognition within a segment. In contrast, localization demands temporally sensitive, fine-grained features to pinpoint the exact moment of transition. The prevailing paradigm to resolve this conflict is task decoupling, which employs separate decoders for each task atop a shared spatio-temporal encoder (*e.g.*, GCNs and TCNs) (Filtjens et al., 2022; Li et al., 2024a; 2021a; Ghosh et al., 2020; Li et al., 2023d;c; Yan et al., 2018; Chai et al., 2024), as shown in Fig. 1(a). However, we argue this popular strategy is a critical over-simplification. While the features may compete, the

---

Corresponding author: Mei Wang (wangmei1@bnu.edu.cn), Xinran Duan contributed equally to this work.

tasks themselves are highly complementary at a semantic level; knowing *what* action is occurring provides powerful priors for *where* its boundaries lie, and vice versa. By isolating the two, current methods create "information silos" that preclude beneficial cross-task synergies and artificially limit performance.

Recent refinements include decoupling spatio-temporal modeling to mitigate feature over-smoothing (Li et al., 2023d) and integrating language priors for enhanced representation learning (Ji et al., 2024) but these improvements do not address the fundamental limitation of insufficient cross-task collaboration. To address this limitation, we introduce Curvature-Guided Task Synergy (CurvSeg) for STAS, which restores the intrinsic synergy between decoupled tasks through two key innovations: curvature-based synergy and adaptive feature extraction, as shown in Fig. 1(b).

Our approach is inspired by a profound geometric insight, recently explored in representation learning (Shin et al., 2024), as shown in Fig. 1(c): in a well-learned feature space, trajectories of a continuous data sequence (like skeleton frames) are spatially confined within their respective class clusters. This confinement naturally induces high curvature throughout an action segment as the feature path continuously turns to avoid crossing the cluster's boundary. Conversely, the path straightens out during transitions between actions, resulting in low-curvature "valleys".Consequently, we first exploit this geometric property for cross-task collaboration in STAS, establishing a synergistic loop between feature learning and temporal localization.The curvature of classification features serves as a geometric prior to guide

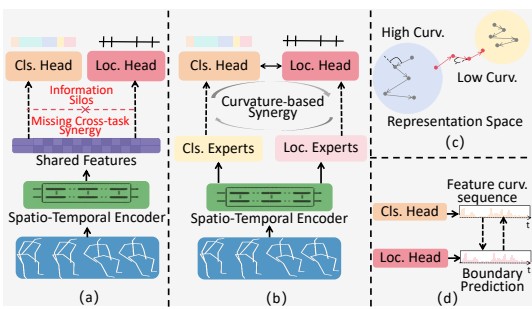

Figure 1: (a) Existing STAS approaches. (b) The proposed method. (c) Classification representations create compact clusters, forcing high trajectory curvature within action segments. (d) Curvature-guided task collaboration.

boundary detection, with curvature valleys indicating likely transition points. Reciprocally, localization predictions supervise the classification feature space by penalizing low curvature within predicted action segments. This symbiotic relationship encourages the generation of more discriminative, compact clusters, which yield more precise geometric priors, creating a synergistic training strategy of improvement.

However, the effectiveness of this curvature-based synergy depends on the quality of the underlying features. Current approaches of using a single, shared encoder output for both tasks creates a compromised representation, failing to optimally serve either classification or localization. To overcome this limitation, we introduce a dual-expert weighting mechanism within a Mixture of Experts (MoE) framework that enables task-adaptive feature extraction from the shared encoder outputs. Specifically, we deploy a separate expert module: the classification expert extracts semantic-oriented representations optimized for action recognition, while the localization expert focuses on fine-grained temporal details essential for boundary detection. This dual-expert mechanism operates across both temporal and spatial dimensions, enabling task-specific feature refinement that optimizes representations for their respective objectives. By providing task-specific feature distillation rather than shared representations, our MoE framework ensures that each decoder head receives optimally tailored inputs, creating an ideal foundation for the curvature-based task collaboration strategy.

The contributions of this work are threefold: First, it proposes a novel curvature-based task synergy mechanism that exploits the geometric properties of feature sequences to establish effective collaboration between decoupled classification and localization sub-tasks. Second, it introduces a dual-expert weighting mechanism within a MoE framework that provides task-adaptive feature extraction, with separate experts extracting semantic representations for classification and fine-grained temporal details for localization to improve the performance of synergy mechanism. Third, comprehensive experiments demonstrate that our curvature-based synergy mechanism effectively enhances action segmentation performance across multiple benchmark datasets.

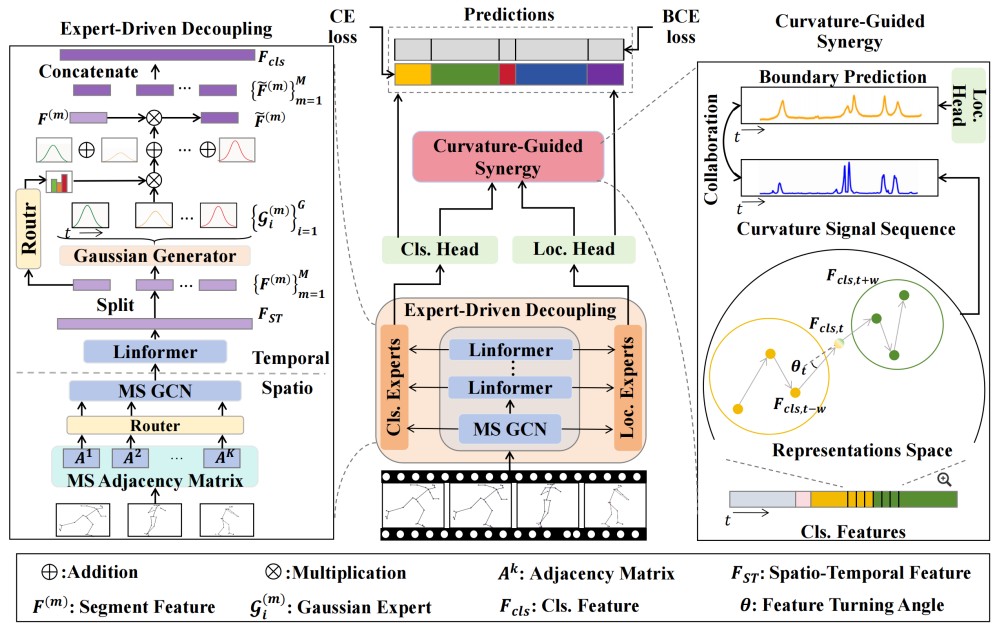

Figure 2: Overview of CurvSeg. Our model processes video features through EDD to capture decoupled classification and localization representations utilizing task-specific experts. Subsequently, CGS leverages geometric curvature principles to guide task collaboration, enhancing both action boundary detection and classification performance.

## 2 RELATED WORK

**Temporal Action Segmentation.** Video-based TAS methods primarily utilize RGB or optical flow features for action understanding. Early RNN-based approaches focused on sequential modeling (Ding & Xu, 2017; Singh et al., 2016), followed by TCN-based methods to capture long-range dependencies (Lea et al., 2017; Farha & Gall, 2019; Li et al., 2020; 2021b). Recent TCN variants have incorporated advanced strategies including receptive field optimization (Gao et al., 2021; Li et al., 2024b; 2025; Gao et al., 2022), boundary-aware mechanisms (Ishikawa et al., 2021; Wang et al., 2020), multi-scale fusion (Singhania et al., 2023), and diffusion models (Liu et al., 2023a). Meanwhile, Transformer-based architectures have emerged as powerful alternatives, leveraging attention mechanisms for adaptive context modeling (Bahrami et al., 2023; Behrmann et al., 2022; Du et al., 2023; Wang et al., 2024; Yi et al., 2021). However, limitations in privacy and robustness to appearance variance hinder video-based methods, necessitating alternative modalities such as skeleton sequence.

**Skeleton-based Temporal Action Segmentation.** STAS methods leverage skeleton data to capture motion dynamics while remaining decoupled from visual appearance. Traditional approaches typically adopt a two-stage pipeline combining Graph Convolutional Networks (GCNs) for spatial modeling and Temporal Convolutional Networks (TCNs) for long-range temporal dependencies (Filtjens et al., 2022; Ghosh et al., 2020). Recently, significant advancements have focused on refining this architecture to handle complex motion patterns. DeST (Li et al., 2023d) decouples spatio-temporal modeling to mitigate feature over-smoothing, while LAC (Yang et al., 2023) explores latent motion primitives. Most recently, LaSA (Ji et al., 2024) integrates language priors for enhanced representation learning. However, these methods primarily address architectural or semantic aspects, often overlooking the intrinsic synergy between tasks. Unlike current decoupled paradigms, we explicitly construct a geometric bridge to restore cross-task collaboration within the feature space.

**Task Decoupling in STAS.** The conflict between classification and regression tasks, initially observed in object detection, has led to a widespread adoption of decoupled classification and localization heads in object detectors (Wu et al., 2020; Ge et al., 2021). Similar to object detection, existing STAS methods (Li et al., 2023d; Ji et al., 2024) employ separate heads for action classification and boundary localization with shared input feature. However, task decoupling paradigm suffers from

two critical limitations: the separate decoder heads fail to enable explicit cross-task collaboration, and the shared encoder with identical input features leads to suboptimal trade-offs between competing task requirements. To address these limitations, we propose a framework that simultaneously maintains task-specific feature adaptation and enables effective synergies through curvature-based geometric collaboration and dual-expert feature enhancement.

## 3 METHOD

### 3.1 PROBLEM SETTING

Given a skeleton sequence $\mathbf{F}_s \in \mathbb{R}^{D_{in} \times T \times V}$ obtained from motion capture systems or pose estimation algorithms, where $D_{in}$, $T$, and $V$ represent the number of channels(where $D_{in}$ is 3 for 3D coordinates), frames, and joints respectively. The goal of skeleton-based temporal action segmentation is to predict the action label for each frame, with ground truth labels $\mathbf{Y} \in \{1, 2, ..., C\}^T$, where $C$ denotes the number of action classes.

### 3.2 FOUNDATION FRAMEWORK

The STAS task can be decomposed into two sub-tasks: action classification and boundary localization. We follow the typical backbone architecture from DeST (Li et al., 2023d) and LaSA (Ji et al., 2024), which employs spatial-temporal modeling with dual decoders for classification and localization.

**Spatial Modeling.** Given an input skeleton sequence $\mathbf{F}_s$, spatial feature $\mathbf{F}_{gcn}$ extraction is performed by a multi-scale graph convolution network (MS-GCN), which captures joint correlations at different receptive fields. For each hop $k$, an adjacency matrix $\mathbf{A}^k \in \{0, 1\}^{V \times V}$ is constructed based on the shortest joint distance, and normalized multi-scale adjacency $\mathbf{A}_{MS} \in \{0, 1\}^{V \times KV}$ is formed by concatenating all $K$ hops. The multi-scale spatial features are computed as:

$$\mathbf{F}_{gcn} = ReLU[(\mathbf{A}_{MS} + \mathbf{B})\mathbf{F}_s\mathbf{W}_s], \tag{1}$$

where $\mathbf{B} \in \mathbb{R}^{V \times (K \cdot V)}$ is a trainable matrix for adaptive joint relationships and $\mathbf{W}_s \in \mathbb{R}^{1 \times 1 \times (K \cdot D) \times D}$ handles channel adjustment. Based on DeST, we integrate the joint-wise skeletal features $\mathbf{F}_{gcn}$ into the global frame-wise feature $\mathbf{F}_{ST}$ through spatio-temporal attention fusion with joint embeddings. This module yields spatially enhanced frame-wise joint representations.

**Temporal Modeling.** For temporal modeling, as in DeST and LaSA, Linear Transformer is adopted to achieve global temporal dependencies with $O(n)$ complexity. The $l$-th linear transformer layer is computed as:

$$\mathbf{F}_{ST}^{l+1} = ReLU[\phi(\mathbf{Q}_t)(\phi(\mathbf{K}_t^\top)\mathbf{V}_t)\mathbf{W}_t + \mathbf{F}_{ST}^l], \tag{2}$$

where $\mathbf{Q}_t$, $\mathbf{K}_t$, $\mathbf{V}_t$ are query, key, value matrices derived from frame feature $\mathbf{F}_{ST} \in \mathbb{R}^{D \times T}$, and $\phi(\cdot)$ denotes the sigmoid activation function.

**Decoder Head.** The final spatio-temporal features $\mathbf{F}_{ST}^L \in \mathbb{R}^{D \times T}$ from the last layer of encoder are fed into two distinct heads to obtain class prediction logits $\hat{\mathbf{Y}}^{cl} \in \mathbb{R}^{C \times T}$ and boundary prediction logits $\hat{\mathbf{Y}}^b \in \mathbb{R}^{1 \times T}$. The predictions are further refined by a segmentation branch (Linear Transformer) and a boundary branch (TCN).

Frame-level cross-entropy loss and segment-level smoothness loss are applied to the class predictions:

$$\mathcal{L}_c = -\frac{1}{T}\sum_{t=1}^{T}\log(\hat{y}_{t,c}^{cl}) + \frac{1}{TC}\sum_{t=1}^{T}\sum_{c=1}^{C}\left[\log\left(\hat{y}_{t-1,c}^{cl}/\hat{y}_{t,c}^{cl}\right)\right]^2, \tag{3}$$

where $\hat{y}_{t,c}^{cl}$ denotes the predicted probability of ground truth label $c$ at time $t$. For boundary localization, binary logistic regression loss is utilized:

$$L_b = -\frac{1}{T}\sum_{t=1}^{T}\left[y_t^b \log\left(\hat{y}_t^b\right) + \left(1 - y_t^b\right)\log\left(1 - \hat{y}_t^b\right)\right], \tag{4}$$

where $y_t^b$ is the boundary ground truth (1 for boundary frames, 0 otherwise).

where

### 3.3 CURVATURE-GUIDED SYNERGY

**Motivation.** Existing STAS methods suffer from limited cross-task collaboration, treating classification and localization as independent processes despite their inherent correlations. To address this limitation, we leverage a fundamental geometric principle from representation learning (Parulekar et al., 2023; Wang et al., 2022b) when classification representations successfully separate different action classes, they confine action sequence trajectories within compact, class-specific regions. As formally derived in Appendix B, this geometric constraint implies that the average curvature of a random walk is inversely proportional to the radius of its bounding hyper-sphere. This spatial confinement creates a fundamental geometric phenomenon: intra-segment points must frequently change direction to remain within their class-specific boundary, resulting in high curvature, while inter-segment points exhibit low curvature as they move between class regions. This geometric principle establishes a natural bridge between classification quality and boundary detection, creating a synergistic relationship that enhances both tasks simultaneously.

**Curvature Calculation.** We define the representational trajectories using the frame-wise classification features $\mathbf{F}_{cls} \in \mathbb{R}^{D \times T}$. Specifically, $\mathbf{F}_{cls}$ is obtained by decoupling the global encoder output $\mathbf{F}_{ST}$ through the Expert Decoupling (EDD) module (refer to Sec. 3.4 for details). Note that since spatial pooling is performed within the encoder, $\mathbf{F}_{cls}$ represents global frame-level features, implying that curvature is computed globally per frame rather than per joint.

For each timestamp $t$, we select three consecutive points $\mathbf{F}_{cls,t-w}$, $\mathbf{F}_{cls,t}$, and $\mathbf{F}_{cls,t+w}$ along the trajectory, where the window size $w$ is set to ensure stability and adaptability across different temporal scales. The turning angle $\theta_t$ between two consecutive difference vectors is computed as:

$$\theta_t = arccos \frac{(\mathbf{F}_{cls,t} - \mathbf{F}_{cls,t-w}) \cdot (\mathbf{F}_{cls,t+w} - \mathbf{F}_{cls,t})}{||\mathbf{F}_{cls,t} - \mathbf{F}_{cls,t-w}|| \cdot ||\mathbf{F}_{cls,t+w} - \mathbf{F}_{cls,t}||}, \tag{5}$$

where $\theta_t \in [0, \pi]$ quantifies the instantaneous direction change at timestamp $t$. The curvature $\kappa_t$ is defined as the turning angle normalized by the sum of difference vectors:

$$\kappa_t = \theta_t / (||\mathbf{F}_{cls,t} - \mathbf{F}_{cls,t-w}|| \cdot ||\mathbf{F}_{cls,t+w} - \mathbf{F}_{cls,t}|| + \epsilon), \tag{6}$$

where $\epsilon$ is set to avoid division by zero. Subsequently, we compute the curvature-based boundary change metric from the obtained curvature sequence. We apply moving average smoothing to the raw curvature sequence to reduce noise and enhance temporal consistency, noted as $\bar{\kappa}$. The smoothed curvature sequence is then normalized to ensure scale invariance $\hat{\kappa}_t = \frac{\bar{\kappa}_t - min(\bar{\kappa}_t)}{max(\bar{\kappa}_t) - min(\bar{\kappa}_t)}$, and we obtain the boundary change metric by inverting the normalized curvature values, $\mathcal{C}_t = 1 - \hat{\kappa}_t$.

**Curvature-Based Task Collaboration.** To enable explicit collaboration between classification and localization tasks, we impose a bidirectional consistency constraint between the boundary prediction probabilities:

$$\mathcal{L}_{curv} = -\frac{1}{T} \sum_{t=1}^{T} MSE(\hat{y}_t^b, \varphi(\mathcal{C}_t)) + MSE(\mathcal{C}_t, \varphi(\hat{y}_t^b)), \tag{7}$$

where $MSE(\cdot, \cdot)$ denotes mean squared error distance and $\varphi(\cdot)$ denotes the gradient stop function. This bidirectional constraint creates a synergistic training strategy where the localization branch aligns with geometric properties from classification features, while classification learning benefits from boundary-aware supervision, resulting in mutually enhanced performance.

### 3.4 EXPERT-DRIVEN DECOUPLING

While our curvature-guided synergy mechanism establishes effective cross-task collaboration, its effectiveness depends critically on the quality of features used for curvature computation. However, both decoder heads currently receive identical shared encoder outputs, which are inherently compromised representations that may not fully serve the distinct requirements of classification and localization tasks. To this end, inspired by recent multi-modal perception tasks (Kim et al., 2025), our approach introduces an expert-driven decoupling mechanism, where the experts process the same encoder features but focus on different task-specific aspects. Specifically, we build two specialized spatio-temporal expert modules: classification experts and localization experts, which adaptively weight and refine encoder features according to their respective task requirements. Then, we use experts specializing in classification tasks as an example to illustrate this. For spatio modeling,

given the skeleton feature sequence $\mathbf{F}_{ST} \in \mathbb{R}^{D \times T \times V}$ in the spatio-temporal encoder, we capture task-relevant spatial features to recalibrate each joint by computing a joint attention vector via an SE-style block(Hu et al., 2018):

$$\mathbf{F}_{ST} = \mathbf{F}_{ST} + \text{Sigmoid}\big(\text{MLP}(\mathbf{z}_{st})\big)\mathbf{F}_{ST} \in \mathbb{R}^{D \times T \times V}, \qquad (8)$$

where $\mathbf{z}_{st}$ is obtained by globally pooling $\mathbf{F}_{ST}$ over its temporal dimension. Then, we apply decoupled spatio-temporal interaction (DSTI) layer(Li et al., 2023d; Ji et al., 2024), yielding the feature map $\mathbf{F}_{ST} \in \mathbb{R}^{D \times T}$. For temporal modeling, we deploy a series of Gaussian experts to extract task-relevant information across the global temporal distribution. Given that untrimmed videos contain multiple action segments, our approach systematically processes temporal information through the following steps: For a $T$-frame video, we first uniformly divide it into $M$ segments, where $M > N$ (the number of action segments in the video). Each segment contains $S = \lfloor T/M \rfloor$ frames, and the $m$-th segment feature $\mathbf{F}^{(m)} \in \mathbb{R}^{D \times S}$ can be divided from the spatio-temporal feature $\mathbf{F}_{ST} \in \mathbb{R}^{D \times T}$. This segment-based approach allows the Gaussian experts to learn relative temporal patterns (e.g., "the start of an event") within a normalized local context, rather than absolute positions in a long video, which significantly simplifies learning and enhances generalization.

For each segment $m$, we generate $G$ Gaussian functions to assign adaptive weights to the $S$ frames within that segment. Each Gaussian expert is parameterized by learnable center $\mu_i^{(m)}$ and variance $\sigma_i^{(m)}$, $\mathcal{G}_i^{(m)} = \mathcal{N}(\mu_i^{(m)}, (\sigma_i^{(m)})^2)$. For simplicity, the normalized center $\mu_i^{(m)}$ and variance $\sigma_i^{(m)}$ are computed by a multilayer perceptron with nonlinear activation layers.

Instead of discrete expert selection, our model uses Gaussian experts as soft temporal masks that adaptively weight their contributions based on task-specific requirements. Then the router assigns routing values for each expert based on segment features $\mathbf{F}^{(m)}$:

$$\tau^{(m)} = \text{Sigmoid}(\text{MLP}(\text{Avg}(\mathbf{F}^{(m)}) \cdot \mathbf{W}^g) \in \mathbb{R}^G, \qquad (9)$$

where $\text{Avg}(\cdot)$ denotes average pooling at temporal dimension, and $\mathbf{W}^g \in \mathbb{R}^{D \times G}$ is a learnable weight matrix that dynamically controls the influence of each expert. Subsequently, outputs from all experts are combined through weighted summation, the integration is expressed as:

$$\tilde{\mathbf{F}}^{(m)} = \sum_{i=1}^{G} \tau_i^{(m)} \mathcal{G}_i^{(m)} \mathbf{F}^{(m)}, \qquad (10)$$

The resulting expert weights enable the model to more effectively emphasize task-relevant information, capturing task-specific insights.

## 3.5 Overall Optimization Objective

Based on the proposed scheme, the overall optimization objective of the STAS model can be designed as follows:

$$\mathcal{L} = \mathcal{L}_c + \mathcal{L}_b + \lambda \mathcal{L}_{curv}, \qquad (11)$$

where $\lambda$ is a hyper-parameter for balance loss functions.

## 4 Experiment

### 4.1 Dataset and Evaluation Metrics

**Datasets.** Following DeST (Li et al., 2023d) and LaSA (Ji et al., 2024), we evaluate on four standard temporal action segmentation datasets: MCFS-22/MCFS-130 (Liu et al., 2021) (22/130 figure skating actions), PKU-MMD (Liu et al., 2017) (52 daily activities with X-sub/X-view splits), and LARa (Niemann et al., 2020) (8 warehouse activities). We follow standard protocols with 5-fold cross-validation for MCFS and single-split evaluation for PKU-MMD and LARa.

**Evaluation Metrics.** We report three standard metrics: (1) Frame-wise Accuracy (Acc), (2) Segmental Edit Score (Edit), and (3) Segmental F1 Score (F1@10, 25, 50) at IoU thresholds of 10%, 25%, and 50%. Segmental metrics provide comprehensive evaluation by penalizing over-segmentation errors.

Table 1: Comparison of methods on PKU-MMD and LARa datasets

| Model | PKU-MMD (X-sub) | | | | | PKU-MMD (X-view) | | | | | LARa | | | | |
|---|---|---|---|---|---|---|---|---|---|---|---|---|---|---|---|
| | Acc | Edit | F1@{10, | 25, | 50} | Acc | Edit | F1@{10, | 25, | 50} | Acc | Edit | F1@{10, | 25, | 50} |
| MS-TCN++ (Li et al., 2023b) | 66.0 | 66.7 | 69.6 | 65.1 | 51.5 | 58.4 | 56.7 | 58.7 | 53.2 | 38.7 | 71.7 | 58.6 | 60.1 | 58.6 | 47.0 |
| ETSN (Li et al., 2021b) | 68.4 | 67.1 | 70.4 | 65.5 | 52.0 | 60.7 | 57.6 | 62.4 | 57.9 | 44.3 | 71.9 | 58.4 | 64.3 | 60.7 | 48.1 |
| ASRF (Ishikawa et al., 2021) | 67.7 | 67.1 | 72.1 | 68.3 | 56.8 | 60.4 | 59.3 | 62.5 | 58.0 | 46.1 | 71.9 | 63.0 | 68.3 | 65.3 | 53.2 |
| MS-GCN (Filtjens et al., 2022) | 70.0 | 65.8 | 68.5 | 63.9 | 50.1 | 66.5 | 64.0 | 67.1 | 62.4 | 49.9 | 73.7 | 58.6 | 63.8 | 59.4 | 47.6 |
| MTST-GCN (Chai et al., 2024) | 70.0 | 65.8 | 68.5 | 63.9 | 50.1 | 66.5 | 64.0 | 67.1 | 62.4 | 49.9 | 73.7 | 58.6 | 63.8 | 59.4 | 47.6 |
| ME-ST (Ji et al., 2025) | 74.1 | 70.5 | 76.6 | 73.2 | 62.4 | 68.5 | 67.2 | 72.3 | 68.8 | 58.1 | 74.2 | 65.0 | 71.0 | 68.2 | 57.1 |
| DeST-TCN (Li et al., 2023d) | 67.6 | 66.3 | 71.7 | 68.0 | 55.5 | 62.4 | 58.2 | 63.2 | 59.2 | 47.6 | 72.6 | 63.7 | 69.7 | 66.7 | 55.8 |
| DeST (Li et al., 2023d) | 70.3 | 69.3 | 74.5 | 71.0 | 58.7 | 67.3 | 64.7 | 69.3 | 65.6 | 52.0 | 75.1 | 64.2 | 70.3 | 68.0 | 57.7 |
| +Ours | 71.8 | 69.9 | 75.4 | 71.8 | 59.2 | 67.9 | 65.4 | 69.9 | 66.3 | 52.4 | 76.1 | 65.6 | 71.7 | 68.8 | 58.7 |
| LaSA (Ji et al., 2024) | 73.5 | 73.4 | 78.3 | 74.8 | 63.6 | 69.5 | 67.8 | 72.9 | 69.2 | 57.0 | 75.3 | 65.7 | 71.6 | 69.0 | 57.9 |
| **+Ours** | **74.3** | **74.4** | **79.3** | **76.3** | **65.5** | **71.0** | **68.9** | **74.4** | **70.7** | **58.0** | **76.6** | **66.2** | **72.5** | **70.0** | **59.0** |

Table 2: Comparison of methods on MCFS-22 and MCFS-130 datasets

| Model | MCFS-22 | | | | | MCFS-130 | | | | |
|---|---|---|---|---|---|---|---|---|---|---|
| | Acc | Edit | F1@{10, | 25, | 50} | Acc | Edit | F1@{10, | 25, | 50} |
| MS-TCN (Farha & Gall, 2019) | 75.6 | 74.2 | 74.3 | 69.7 | 59.5 | 65.7 | 54.5 | 56.4 | 52.2 | 42.5 |
| ASTFormer (Yi et al., 2021) | 78.7 | 82.3 | 82.8 | 77.9 | 66.9 | 67.5 | 69.1 | 68.3 | 64.0 | 55.1 |
| MS-GCN (Filtjens et al., 2022) | 75.5 | 72.6 | 75.7 | 70.5 | 57.9 | 64.9 | 52.6 | 52.4 | 48.8 | 39.0 |
| ID-GCN+ASRF (Filtjens et al., 2022) | 78.1 | 81.6 | 86.4 | 83.4 | 73.0 | 67.1 | 68.2 | 68.7 | 65.6 | 56.9 |
| DeST (Li et al., 2023d) | 80.4 | 85.2 | 87.4 | 84.5 | 75.0 | 71.4 | 75.8 | 75.8 | 72.2 | 63.0 |
| LaSA (Ji et al., 2024) | 80.8 | 86.7 | 89.3 | 86.2 | 76.3 | 72.6 | 79.3 | 79.3 | 75.8 | 66.6 |
| **Ours** | **81.2** | **87.7** | **89.7** | **86.6** | **76.7** | **73.1** | **79.8** | **80.0** | **76.6** | **66.7** |

## 4.2 IMPLEMENTATION DETAILS

All experiments are conducted on a single 3090 GPU with Adam optimizer. We set batch size to 1 and learning rate to 0.0005 for MCFS-22/130, training for 300 epochs. For PKU-MMD and LARa datasets, we use batch size 4 and 3 with learning rate 0.001, training for 300 and 40 epochs respectively. Besides, we use a temporal window size of $w = 10$ for curvature computation. The loss balancing coefficient $\lambda$ is set to 4 for the PKU-MMD dataset, 2.5 for the LARa dataset, and 2 for the MCFS datasets. In the temporal Mixture-of-Experts network, each video is divided into 64 segments, and 2 Gaussian experts are generated for each segment.

## 4.3 COMPARISONS WITH THE STATE-OF-THE-ART.

We benchmark our method against state-of-the-art approaches, surpassing current methods' performance across all evaluated datasets (Tab. 1 and Tab. 2). Notably, the most substantial gains are observed in segmental F1 scores, directly validating that our method enhances temporal boundary precision. Crucially, this is not a one-way benefit. The concurrent improvement in frame-wise accuracy demonstrates the reciprocal nature of our design: refining boundaries leads to purer feature representations, which in turn benefits the classification task, confirming a truly synergistic loop.

Table 3: Ablation Study Results on PKU and LARa dataset

| Model | PKU-MMD (X-sub) | | | | | PKU-MMD (X-view) | | | | | LARa | | | | |
|---|---|---|---|---|---|---|---|---|---|---|---|---|---|---|---|
| | Acc | Edit | F1@{10, | 25, | 50} | Acc | Edit | F1@{10, | 25, | 50} | Acc | Edit | F1@{10, | 25, | 50} |
| base | 73.5 | 73.4 | 78.3 | 74.8 | 63.6 | 69.5 | 67.8 | 72.9 | 69.2 | 57.0 | 75.3 | 65.7 | 71.6 | 69.0 | 57.9 |
| CGS | 74.0 | 74.3 | 78.7 | 76.0 | 64.7 | 70.9 | **69.1** | **74.5** | 70.3 | 57.6 | 76.2 | 65.9 | 72.1 | 69.4 | 58.7 |
| EDD | 73.8 | 73.8 | 78.3 | 75.3 | 64.0 | 69.9 | 68.2 | 74.2 | 69.6 | 57.4 | 76.2 | 65.8 | 72.0 | 69.4 | 58.4 |
| Ours | **74.3** | **74.4** | **79.3** | **76.3** | **65.5** | **71.0** | 68.9 | 74.4 | **70.7** | **58.0** | **76.6** | **66.2** | **72.5** | **70.0** | **59.0** |

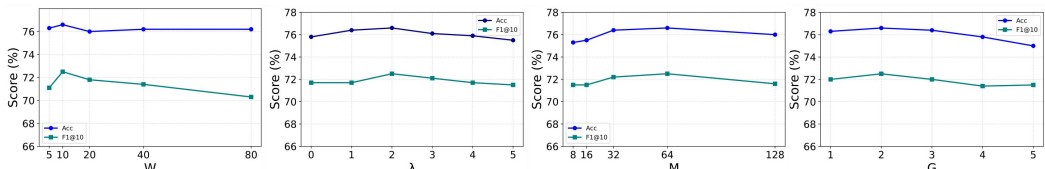

Figure 3: Hyper-parameters analysis. '$w$': window size; '$\lambda$': the CGS loss weigt; '$M$':the granularity of EDD analysis;'$G$': the number of Gaussian experts.

## 4.4 Ablation Studies

We conduct ablation studies on the LaSA Ji et al. (2024) baseline to dissect the contribution of our two core components: Expert-Driven Decoupling (EDD) and Curvature-Guided Synergy (CGS). As shown in Tab. 3, each component is both independently effective and mutually reinforcing.

**Effectiveness of Expert-Driven Decoupling (EDD).** Adding only the EDD module (base+EDD) yields a consistent performance boost across all metrics. This confirms our hypothesis that providing specialized features via task experts creates a superior foundation for both classification and localization, overcoming the inherent compromises of a shared feature representation.

Table 4: CGS ablation studies on LARa dataset.

| Metric | Acc | Edit | F1@{10, | 25, | 50} |
|---|---|---|---|---|---|
| Base | 75.3 | 65.7 | 71.6 | 69.0 | 57.9 |
| Euclid | 76.0 | 65.1 | 71.7 | 68.9 | 57.8 |
| Cosine | 75.2 | 65.2 | 71.5 | 68.6 | 57.0 |
| Grad. Saliency | 74.4 | 64.3 | 70.7 | 67.9 | 57.1 |
| **Curv** | **76.2** | **65.9** | **72.1** | **69.4** | **58.7** |

**Effectiveness of Curvature-Guided Synergy (CGS).** Independently adding the CGS module (base+CGS) brings the most substantial gains to the segmental F1 scores. This directly validates that our curvature-guided loop is highly effective at enhancing boundary precision. The concurrent rise in accuracy further demonstrates the mechanism's reciprocal benefit: better boundary priors lead to better-clustered features for classification.

**Synergy of the Full Model.** Finally, our full model, combining both components, achieves the best results and demonstrates a clear synergistic effect—its performance gain significantly surpasses the simple sum of the individual modules' contributions. This validates our core design philosophy: EDD provides the high-quality, specialized features that act as an optimal foundation, allowing the CGS module to then leverage these refined inputs to its full geometric potential for precise segmentation. We note that on PKU-MMD (X-view), CGS-only marginally leads in Edit and loose F1@10 scores. This implies that under drastic view shifts, the smoother shared representations of CGS favor prediction continuity. However, the full model dominates on strict metrics (F1@{25, 50}) and Accuracy, verifying that EDD's specialized features are indispensable for **sharpening boundaries** and distinguishing fine-grained semantics.

## 4.5 Component Analysis

### 4.5.1 Curvature-Guided Synergy Analysis.

**Analysis of Synergy.** Our ablation study (Tab. 6) dissects the CGS module within the overall framework. Isolating the forward path (C→L) primarily boosts F1 scores by refining boundaries, while the backward path (L→C) enhances accuracy by regularizing features. Crucially, the full CGS module yields a synergistic gain significantly larger than the sum of its parts, confirming that boundary guidance and feature refinement are mutually reinforcing.

**Superiority of Curvature-based Guidance.** We validate our approach against traditional distance metrics (Euclidean, Cosine) and learned boundary detectors ( *e.g.* gradient-based saliency) in Table 4. Results demonstrate that curvature consistently provides the best guidance. Existing baselines show limitations: distance metrics are sensitive to feature magnitude, while saliency maps tend to highlight *discriminative* action centers (e.g., peak motion) rather than temporal boundaries. In contrast, curvature explicitly captures the **directional evolution** of the feature manifold. This geometric

prior serves as a precise, parameter-free proxy for state transitions, remaining robust to both subtle drifts and abrupt changes.

**Curvature as a Direct Boundary Proxy.** To further validate our premise, we use the inverted curvature values directly as boundary predictions on Lara dataset. This simple threshold-based method achieved remarkably as shown in Tab. 12. This strongly confirms that low-curvature points in the feature trajectory are a highly effective proxy for temporal boundaries, validating the foundational principle of our CGS module.

Table 5: Refine with different boundary prediction.

| Boundary | Acc | Edit | F1@{10, | 25, | 50} |
|---|---|---|---|---|---|
| Pred | **76.6** | **66.2** | **72.5** | **70.0** | **59.0** |
| Curv | 75.4 | 64.6 | 72.3 | 69.4 | 55.1 |

Table 6: Task synergy ablation study on LARa

| Metric | Acc | Edit | F1@{10, | 25, | 50} |
|---|---|---|---|---|---|
| Base | 75.3 | 65.7 | 71.6 | 69.0 | 57.9 |
| C→L | 76.1 | 65.3 | 71.7 | 68.8 | 58.2 |
| L→C | 75.9 | 65.4 | 71.6 | 69.1 | 58.1 |
| **CGS** | **76.6** | **66.2** | **72.5** | **70.0** | **59.0** |

Table 7: EDD ablation study on LARa dataset.

| Metric | Acc | Edit | F1@{10, | 25, | 50} |
|---|---|---|---|---|---|
| Base | 75.3 | 65.7 | 71.6 | 69.0 | 57.9 |
| Indep. Enc. | 75.7 | 64.5 | 71.6 | 68.3 | 57.7 |
| Pyr. Decoupling | 75.6 | 64.7 | 71.2 | 68.4 | 57.9 |
| **EDD** | **76.2** | **65.8** | **72.0** | **69.4** | **58.4** |

### 4.5.2 EXPERT-DRIVEN DECOUPLING ANALYSIS

**Advantage of Dynamic Decoupling.** We benchmarked our EDD against alternative decoupling strategies in Tab. 7 on Lara dataset. Our approach is superior because it avoids two common pitfalls: it neither compromises features through forced sharing (Base model) nor loses valuable signals through complete isolation (Indep. Enc.). Unlike static allocation (Pyr. Decoupling), our expert routing mechanism dynamically creates specialized features tailored to each task's needs. This allows the model to achieve the best of both worlds: generating highly discriminative, task-specific representations while still enabling beneficial, selective information exchange.

### 4.6 HYPERPARAMETER ANALYSIS

We conduct a brief analysis on key hyper-parameters in Fig. 3, including curvature window size $w$, the CGS loss weight $\lambda$, the granularity of EDD analysis $M$, and the number of Gaussian experts $G$.

The hyperparameter $\lambda$ balances the contribution of our Curvature Guidance Synergy (CGS) loss. Empirically, the performance peaks at $\lambda=2$. A smaller value provides insufficient guidance, while a larger value overly constrains the model and hurts performance. Segment Count $M$ controls the temporal resolution. The model achieves the best results with $M = 64$, striking an optimal balance. A smaller $M$ offers poor temporal precision, whereas a larger M causes context loss within each short segment. Gaussian Expert Count $G$ sets the modeling capacity within each segment. Performance is optimal at $G = 2$, while more experts ($G > 2$) lead to diminishing returns and potential overfitting. Hyper-parameter $w$ controls the temporal receptive field. We found $w=10$ optimal, balancing noise suppression and boundary sensitivity. Smaller windows ($w=5$) capture insufficient context, while larger ones ($w >= 40$) over-smooth the sharp directional changes that define action boundaries. Detailed hyperparameter analysis is provided in Appendix C.

### 4.7 ERROR ANALYSIS AND LIMITATIONS

Our analysis identifies two scenarios where geometric priors face intrinsic challenges. First, **low-dynamic actions** (e.g., *Step Sequences*) exhibit continuous feature evolution without abrupt shifts. This results in shallower curvature valleys compared to high-dynamic actions (e.g., *Jumps*), making boundary localization physically more ambiguous Additionally, skeleton sequences inevitably contain sensor noise or estimation jitter. This can introduce high-frequency fluctuations in the feature trajectory, creating spurious curvature peaks (false positives) in non-boundary regions, as shown in Fig. 4. However, despite these geometric constraints, quantitative results (Tab. 8) demonstrate that our method remains highly robust. Notably, we achieve a larger relative improvement over the

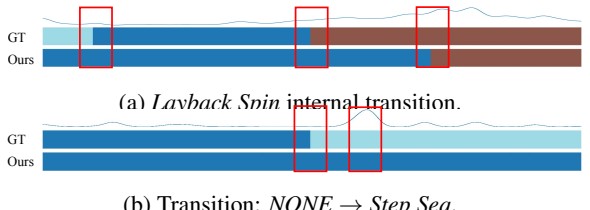

(a) *Layback Spin* internal transition.

(b) Transition: *NONE → Step Seq.*

Figure 4: **Visualization of Limitation Cases.** (a) Internal spin variations cause noise. (b) Gradual step transitions lack deep curvature valleys.

| Category | Metric | Base | Ours |
|---|---|---|---|
| High-Dynamic | F1@10 | 71.18 | **72.86** |
| | Edit | 58.44 | **61.61** |
| Low-Dynamic | F1@10 | 49.62 | **52.86** |
| | Edit | 21.81 | **27.83** |

Table 8: **Performance on High vs. Low Dynamic Actions.** Low-dynamic actions show lower absolute scores but higher relative gains from our method.

baseline on these challenging low-dynamic actions (+6.02% Edit Score) compared to high-dynamic ones (+3.17%). This indicates that even weak curvature signals act as effective differential cues, providing structural constraints that guide the model to resolve ambiguities where pure semantic baselines typically fail.

### 4.8 QUALITATIVE COMPARISON

Fig. 5 shows qualitative comparisons with previous methods, the top row shows the curvature visualization results of the skeleton sequence features. Our approach utilizes curvature-based task collaboration and Expert-driven adaptive feature extraction to achieve more accurate action prediction and boundary detection. The curvature mechanism enables effective cross-task synergy through geometric feature properties, while specialized experts provide task-specific representations for classification and localization. These improvements result in segmentation outputs that closely match ground truth annotations.

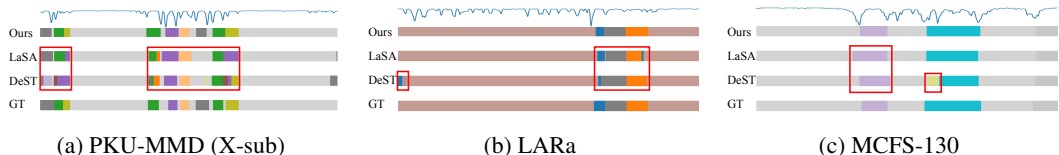

(a) PKU-MMD (X-sub)    (b) LARa    (c) MCFS-130

Figure 5: Qualitative comparison of segmentation results on PKU-MMD (X-sub), LARa, MCFS-130. Our method alleviates **over-segmentation** and achieves more precise action boundary localization, demonstrating better temporal consistency and robustness compared to baseline methods.

## 5 CONCLUSION

In this work, we propose a novel framework for skeleton-based temporal action segmentation that addresses the fundamental limitation of insufficient cross-task collaboration. Our approach introduces a curvature-based task synergy mechanism that exploits geometric properties of feature sequences to establish effective collaboration between classification and localization sub-tasks, coupled with a dual-expert weighting mechanism within a Mixture of Experts framework for task-adaptive feature extraction. However, our method still faces limitations in handling extremely noisy skeleton data and complex multi-person scenarios, leaving room for further improvement. Future work should explore more robust geometric priors and investigate the integration of multi-modal information to enhance the framework's applicability in real-world deployment scenarios. See Appendix D for reproducibility statement and Appendix E for our statement on LLM usage.

### ACKNOWLEDGMENTS

The authors gratefully acknowledge the financial support from the National Natural Science Foundation of China (NSFC). This work was supported by NSFC under Grant Nos. 62437001 and 62506040.

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

---

**Algorithm 1** Curvature-Guided Synergy (CGS)

---

**Require:** Mini-batch $\mathcal{B} = \{(\mathbf{X}, \mathbf{Y}, \mathbf{y}^b)\}$; encoder/backbone $\mathcal{E}$; classification head $\mathcal{H}_{cls}$; boundary head $\mathcal{H}_b$; window size $w$; smoothing window $m$; weight $\lambda$; small $\epsilon$; stop-gradient $\varphi(\cdot)$
**Ensure:** CGS loss $\mathcal{L}_{\text{curv}}$ and total loss $\mathcal{L}_{\text{total}}$

1: **for** each sequence $(\mathbf{X}, \mathbf{Y}, \mathbf{y}^b) \in \mathcal{B}$ **do**
2:      $\mathbf{F}_{ST} \leftarrow \mathcal{E}(\mathbf{X})$                                 ▷ Spatial-temporal encoding (Sec. Foundation)
3:      $\hat{\mathbf{Y}}^{cl} \leftarrow \mathcal{H}_{cls}(\mathbf{F}_{ST}), \quad \hat{\mathbf{y}}^b \leftarrow \mathcal{H}_b(\mathbf{F}_{ST})$                       ▷ Dual heads
4:      $\mathbf{F}_{cls} \leftarrow$ frame-wise classification features from encoder/class head

5:      **Curvature construction on classification trajectory**
6:      **for** $t = w, \ldots, T - w$ **do**
7:          $\mathbf{u}_t \leftarrow \mathbf{F}_{cls,t} - \mathbf{F}_{cls,t-w}; \quad \mathbf{v}_t \leftarrow \mathbf{F}_{cls,t+w} - \mathbf{F}_{cls,t}$
8:          $\theta_t \leftarrow \arccos\left(\dfrac{\langle \mathbf{u}_t, \mathbf{v}_t \rangle}{\|\mathbf{u}_t\| \, \|\mathbf{v}_t\| + \epsilon}\right)$                     ▷ Turning angle, cf. Eq. (7)
9:          $\kappa_t \leftarrow \dfrac{\theta_t}{\|\mathbf{u}_t\| \, \|\mathbf{v}_t\| + \epsilon}$                            ▷ Curvature, cf. Eq. (8)
10:     **end for**
11:     $\bar{\kappa} \leftarrow \text{MovingAverage}(\kappa, m)$                           ▷ Noise suppression
12:     $\hat{\kappa}_t \leftarrow \dfrac{\bar{\kappa}_t - \min(\bar{\kappa})}{\max(\bar{\kappa}) - \min(\bar{\kappa}) + \epsilon}$             ▷ Min–max normalization
13:     $\mathcal{C}_t \leftarrow 1 - \hat{\kappa}_t$                      ▷ Boundary-change metric from curvature

14:     **Losses**
15:     $\mathcal{L}_c \leftarrow$ frame CE + segment smoothness (Eq. (5))
16:     $\mathcal{L}_b \leftarrow$ binary logistic loss for boundaries (Eq. (6))
17:     $\mathcal{L}_{\text{curv}} \leftarrow \dfrac{1}{T} \sum_{t=1}^{T} \left[ \text{MSE}(\hat{y}_t^b, \varphi(\mathcal{C}_t)) + \text{MSE}(\mathcal{C}_t, \varphi(\hat{y}_t^b)) \right]$     ▷ Bidirectional consistency, cf. Eq. (9)
18: **end for**
19: $\mathcal{L}_{\text{total}} \leftarrow \mathcal{L}_c + \mathcal{L}_b + \lambda \, \mathcal{L}_{\text{curv}}$
20: **return** $\mathcal{L}_{\text{curv}}, \mathcal{L}_{\text{total}}$

---

# A    APPENDIX: ALGORITHM OF THE PROPOSED GURVSEG

## A.1    CURVATURE-GUIDED SYNERGY

Algorithm 1 converts the frame-wise classification features into a geometry-derived boundary prior and enforces bidirectional consistency with the boundary head. Specifically, we (1) build a temporal trajectory from classification features, (2) compute a k-step turning-angle curvature per frame, followed by smoothing and min–max normalization to obtain a boundary-change metric, and (3) align this metric with the predicted boundary probabilities using a stop-gradient operator in both directions. The resulting curvature-based consistency loss is combined with the standard classification and boundary losses, yielding an self-supervised signal that couples classification quality and boundary localization while preserving model plasticity.

## A.2    EXPERT-DRIVEN DECOUPLING

Algorithm 2 disentangles task-specific evidence for classification and localization while sharing a common encoder. Two specialist expert stacks refine the encoder outputs in a complementary manner: (i) a spatial recalibration module applies SE-style joint attention to emphasize task-relevant joints and then uses a decoupled spatio-temporal interaction (DSTI) layer to form sequence features; (ii) a temporal mixture of Gaussian experts operates on uniformly partitioned segments of the sequence, where each expert produces a soft temporal mask parameterized by learnable centers and variances. A lightweight router predicts routing scores for all experts from segment descriptors, and the masked expert outputs are aggregated by a weighted sum. The resulting expert-refined features

---

**Algorithm 2** Expert-Driven Decoupling (EDD)

---

**Require:** Encoder spatial features $\mathbf{F}_{s,t} \in \mathbb{R}^{V \times T}$; segment count $M$; experts $G$; small $\epsilon$
**Ensure:** Expert-refined sequence $\tilde{\mathbf{F}}_{ST} \in \mathbb{R}^{D \times T}$ for a target head (cls or loc)
1: **Spatial recalibration (SE-style) and DSTI**
2: $\quad \mathbf{z}_s \leftarrow \text{GAP}_t(\mathbf{F}_{s,t})$             ▷ Global average over time
3: $\quad \mathbf{a} \leftarrow \sigma(\text{MLP}(\mathbf{z}_s))$              ▷ $\sigma$ is Sigmoid
4: $\quad \hat{\mathbf{F}}_{s,t} \leftarrow \mathbf{F}_{s,t} + \mathbf{a} \odot \mathbf{F}_{s,t}$           ▷ Joint-wise attention
5: $\quad \mathbf{F}_{ST} \leftarrow \text{DSTI}(\hat{\mathbf{F}}_{s,t})$            ▷ $\mathbf{F}_{ST} \in \mathbb{R}^{D \times T}$

6: **Temporal Gaussian experts on uniform segments**
7: $\quad S \leftarrow \lfloor T/M \rfloor$;   partition $\mathbf{F}_{ST}$ into $\{\mathbf{F}^{(m)} \in \mathbb{R}^{D \times S}\}_{m=1}^{M}$
8: **for** $m = 1$ **to** $M$ **do**
9:    $\bar{\mathbf{f}}^{(m)} \leftarrow \text{Avg}_t(\mathbf{F}^{(m)}) \in \mathbb{R}^D$
10:    **for** $i = 1$ **to** $G$ **do**
11:     $(\mu_i^{(m)}, \sigma_i^{(m)}) \leftarrow \text{MLP}_i(\bar{\mathbf{f}}^{(m)})$       ▷ Centers and spreads
12:     $\sigma_i^{(m)} \leftarrow \text{Softplus}(\sigma_i^{(m)}) + \epsilon$       ▷ Ensure positivity
13:     $g_i^{(m)}[s] \leftarrow \exp\big(-\frac{(s-\mu_i^{(m)})^2}{2(\sigma_i^{(m)})^2}\big)$ for $s = 1 \dots S$
14:     $g_i^{(m)} \leftarrow g_i^{(m)}/\big(\sum_{s=1}^{S} g_i^{(m)}[s] + \epsilon\big)$     ▷ Normalize mask
15:     $\mathbf{H}_i^{(m)} \leftarrow \mathbf{F}^{(m)} \odot g_i^{(m)}$        ▷ Time-wise weighted features
16:    **end for**
17:    $\boldsymbol{\tau}^{(m)} \leftarrow \sigma\big(\text{MLP}(\bar{\mathbf{f}}^{(m)})\mathbf{W}^g\big) \in \mathbb{R}^G$      ▷ Routing scores
18:    $\tilde{\mathbf{F}}^{(m)} \leftarrow \sum_{i=1}^{G} \tau_i^{(m)} \mathbf{H}_i^{(m)}$      ▷ Soft aggregation across experts
19: **end for**
20: $\tilde{\mathbf{F}}_{ST} \leftarrow \text{Concat}_m(\tilde{\mathbf{F}}^{(m)})$        ▷ Back to length-$T$ sequence
21: **return** $\tilde{\mathbf{F}}_{ST}$

---

emphasize relative temporal patterns within segments (e.g., onset, middle, end) rather than absolute timestamps, yielding stable gradients, better generalization, and clean decoupling for the two heads.

# B   APPENDIX: THEORETICAL SUPPLEMENT

## B.1   REPRESENTATIONAL TRAJECTORIES

A trajectory is the path an object in motion follows through space and time (Lee et al., 2007). In feature spaces, a representation trajectory $\mathcal{F}_{\mathcal{T}} = \{\mathbf{f}_t\}_{t=1}^{\mathcal{T}}, \{\mathbf{f}_t\} \in \mathbb{R}^D$ in feature space is a curve defined by sequential representations over time. Curvature at a point measures the instantaneous rate of directional change, quantifying how much a curve bends from a straight line.

For three consecutive time points on a trajectory in feature space, $t-1, t, t+1$, their corresponding representations are respectively $\mathbf{f}_{t-1}, \mathbf{f}_{t-1}, \mathbf{f}_{t+1}$. Naturally, we can obtain two difference vectors, $\mathbf{f}_t - \mathbf{f}_{t-1}$ and $\mathbf{f}_{t+1} - \mathbf{f}_t$. he turning angle $\theta_t$ between the two vectors is calculated similarly to Eq. 12:

$$\theta_{ft} = arccos\frac{(\mathbf{f}_t - \mathbf{f}_{t-1}) \cdot (\mathbf{f}_{t+1} - \mathbf{f}_t)}{||\mathbf{f}_t - \mathbf{f}_{t-1}|| \cdot ||\mathbf{f}_{t+1} - \mathbf{f}_t||}, \tag{12}$$

Each turning angle $\theta_t$ lies in $[0, \pi]$. Thus, curvature is the rate at which the difference vector rotates per unit length—*i.e.*, the rate of change of direction between two difference vectors—and is given by Eq. 13:

$$\kappa_{ft} = \frac{\theta_{ft}}{(||\mathbf{f}_t - \mathbf{f}_{t-1}|| \cdot ||\mathbf{f}_{t+1} - \mathbf{f}_t|| + \epsilon)}. \tag{13}$$

## B.2 TRAJECTORY CURVATURE AND GEOMETRIC CONSTRAINTS

For an action subsequence belonging to a specific category $C_i$, its representation trajectory can be assumed to be constrained within a hyper-sphere $S_{C_i}$ of radius $R_i$. This assumption stems from a widely confirmed observation: Representation learning aims to construct a feature space with good class separability. The learning process naturally clusters samples of the same class together, resulting in distinct and compact class clusters. Well-embedded points within a specific category (or segment) can be extracted from their category-specific sphere (Wang et al., 2022a). In other words, the representation trajectory of intra-segment points can be constrained within a category sphere, while the representation trajectory of inter-segment points cannot.

Consider a transition from a class $C_i$ to another class $C_j$. Let $\mathcal{S}_{C_j}$ be the other hyper-spheres of class $C_j$ with radius $R_j$. To simultaneously contain $\mathcal{S}_{C_i}$, $\mathcal{S}_{C_j}$, and the transition points between them, we define a larger hyper-sphere with radius $R_{inter}$. By definition, it necessarily holds that $R_i < R_{inter}$ and $R_j < R_{inter}$.

Based on the temporal coherence inherent in time series (Shin et al., 2022; 2023),a simplifying assumption is introduced: A representation trajectory $\mathcal{F}_{\mathcal{T}}$ is a Markov chain, the generation of the next point $\mathbf{f}_t$ satisfies two conditions: (1) Equal-step property: $\mathbf{f}_t$ is sampled uniformly at random from the unit hyper-sphere centered at $\mathbf{f}_{t-1}$; (2) Constrained-boundary property The entire trajectory is contained within a larger hyper-sphere of radius: $\|\mathbf{f}_{t-1}\| < R$.

The significance of this assumption is that the equal-step-size property (Condition 1) greatly simplifies the calculation of curvature. According to Eq. 13, when the step size (the denominator) is constantly 1, the curvature of the trajectory is equivalent to its turning angle $\theta_t$. This allows us to focus on analyzing the expected value of the turning angle.

Based on the above model, we can define the mean curvature of a trajectory $\kappa_{\mathcal{F}_{\mathcal{T}}}(R)$ under specific constraints:

$$\kappa_{\mathcal{F}_{\mathcal{T}}}(R) = \frac{1}{\|\mathcal{T}\|} \sum_{t \in \mathcal{T}} \mathbb{E}_{\mathbf{f}_t|R}[\theta_t], \tag{14}$$

where $\mathbb{E}_{\mathbf{f}_t|R}[\theta_t]$ is the expectation of the curvature at timestamp $t$ with respect to the distribution of the representations in the confining hyper-sphere of radius $R$.

In a more intuitive way, we analyze a special case in a 2-dimensional space. We define a simplified model, $\mathbf{f}_t$ is an arbitrary point on the border of a confining circle $\mathcal{S}$. In this case, Eq. 14 can be simplified as:

$$\kappa_{\mathcal{F}_{\mathcal{T}}}(R) = \pi - \frac{1}{2} arccos \frac{1}{2R} \tag{15}$$

Specifically, Let the confining circle be centered at the origin $\mathcal{O}$ with radius $R > 1$, and let $\mathbf{f}_t$ lie on its boundary, $\|\mathbf{f}_t\| = R$. Assume unit step size $\|\mathbf{f}_t - \mathbf{f}_{t-1}\| = \|\mathbf{f}_{t+1} - \mathbf{f}_t\| = 1$, and choose $\mathbf{f}_{t-1}$ on the line segment from $O$ to $\mathbf{f}_t$ so that the incoming direction aligns with the radial direction. The next point $\mathbf{f}_{t+1}$ is sampled uniformly on the unit circle centered at $\mathbf{f}_t$, restricted to lie inside the confining circle; equivalently, $\mathbf{f}_{t+1}$ lies on the intersection arc of the two circles.

With unit step size, the discrete curvature equals the turning angle $\theta_t \in [0, \pi]$. Let $\mathcal{C} = \mathbf{f}_t$ and consider triangle $\triangle \mathcal{OCP}$ with $\mathcal{P} = \mathbf{f}_{t+1}$ at the circle intersection. Then $\|\mathcal{OC}\| = R$, $\quad \|\mathcal{CP}\| = 1$, $\quad \|\mathcal{OP}\| = R$,

$$\cos \psi = \frac{R^2 + 1 - R^2}{2R \cdot 1} = \frac{1}{2R} \quad \Rightarrow \quad \psi_{\max} = \arccos\left(\frac{1}{2R}\right). \tag{16}$$

and by the cosine rule, for $\psi = \angle \mathcal{OCP}$, Geometrically, the turning angle and $\psi$ satisfy $\theta = \pi - \psi$, hence the feasible range is $\theta \in \left[\pi - \arccos\left(\frac{1}{2R}\right), \pi\right]$.

Because $\mathbf{f}_{t+1}$ is uniform along the intersection arc, $\psi$ is uniform on $[0, \psi_{\max}]$, and thus $\theta$ is uniform on $[\pi - \psi_{\max}, \pi]$. Therefore, the average curvature (average turning angle) is

$$\kappa_{\mathcal{F}_{\mathcal{T}}}(R) = \mathbb{E}[\theta] = \pi - \frac{1}{2} \arccos\left(\frac{1}{2R}\right). \tag{17}$$

Moreover,

$$\frac{d}{dR}\kappa_{\mathcal{F}_{\mathcal{T}}}(R) = -\frac{1}{2R\sqrt{4R^2-1}} < 0 \quad (R > 1), \tag{18}$$

so $\kappa_{\mathcal{F}_{\mathcal{T}}}(R)$ decreases monotonically with $R$.

Thus, when $\mathbf{f}_t$ lies on the boundary of the smaller constraining circle, the mean curvature at $\mathbf{f}_t$ is higher. As $R$ increases, the random walk has more space, so the trajectory is less likely to hit the boundary. Consequently, when the trajectory remains inside the smaller circle, its mean curvature decreases. The above conclusion can be generalized to higher dimensions, and the proof of the conclusion is provided by Diao et al. (2013).

A constraint circle's radius for an inter-segment trajectory is always larger than that for an intra-segment trajectory, because an inter-segment trajectory passes through two different quasi-spheres. Since $\kappa_{\mathcal{F}_{\mathcal{T}}}(R)$ is a decreasing function, $\kappa_{\mathcal{F}_{\mathcal{T}}}(R_{intra}) > \kappa_{\mathcal{F}_{\mathcal{T}}}(R_{inter})$.

## C APPENDIX: ADDITIONAL EXPERIMENTS

This section analyzes the impact of loss weight $\lambda$, number of segments $M$, number of Gaussian experts $G$, and window size $w$ on performance metrics based on the four tables in the graph (Tab. 9-12). Except for the hyperparameter being examined, all other settings are fixed at their default values.

Hyper-Parameter $\lambda$ balances the curvature/boundary-aware regularization against the primary objective. Increasing $\lambda$ from 0 to 2 consistently improves all metrics; further increasing to $\lambda \geq 3$ yields slight degradation, indicating over-smoothing. At $\lambda = 2$ we obtain the best results: Acc 76.6, Edit 66.2, F1@10/25/50 $= 72.5/70.0/59.0$. A moderate curvature weight strengthens boundary contrast while preserving intra-segment details; too large values over-regularize fine structures.

Hyper-Parameter $M$ controls the segment-level resolution (e.g., codebook/partition capacity). Performance improves steadily from $M = 8$ to $M = 64$ and slightly drops at $M = 128$. The best configuration is $M = 64$ with the same top-line metrics as above. Small $M$ underfits (insufficient capacity, mixing semantics); excessively large $M$ increases variance (data sparsity per segment). $M = 64$ strikes a favorable capacity–stability balance. Besides, an adaptive strategy that dynamically scales $M$ based on action density yields only marginal improvement at the cost of significantly higher computational complexity.

Hyper-Parameter $G$ sets the complexity of the Mixture-of-Gaussians expert module to model multimodal transitions. Moving from $G = 1$ to $G = 2$ yields consistent improvements, while $G \geq 3$ provides diminishing returns and slight drops. The best result is at $G = 2$. Two experts suffice to capture the dominant transition modes given the dataset size; larger $G$ introduces redundancy and less stable estimation.

Hyper-Parameter $w$ defines the temporal context for curvature estimation/smoothing or short-range attention. $w = 10$ is optimal; a too small window ($w = 5$) lacks context, while large windows ($w \geq 40$) over-smooth and blunt boundary peaks. Boundaries typically manifest as short, sharp directional changes. A moderate window suppresses intra-segment noise without washing out changepoints.

Table 9: Effect of hyperparameter $\lambda$ on model performance

| $\lambda$ | Acc | Edit | F1@{10, 25, 50} | | |
|---|---|---|---|---|---|
| 0 | 75.8 | 65.3 | 71.7 | 69.0 | 57.4 |
| 1 | 76.4 | 65.2 | 71.7 | 69.3 | 57.8 |
| 2 | **76.6** | **66.2** | **72.5** | **70.0** | **59.0** |
| 3 | 76.1 | 65.5 | 72.1 | 69.6 | 58.1 |
| 4 | 75.9 | 64.8 | 71.7 | 68.5 | 57.9 |

Table 10: Effect of segment count $M$ on model performance

| $M$ | Acc | Edit | F1@{10, 25, 50} | | |
|---|---|---|---|---|---|
| 8 | 76.3 | 64.9 | 71.5 | 68.3 | 56.7 |
| 16 | 75.5 | 64.6 | 71.5 | 68.4 | 56.8 |
| 32 | 76.4 | 65.9 | 72.2 | 69.7 | 58.3 |
| 64 | **76.6** | **66.2** | **72.5** | **70.0** | **59.0** |
| 128 | 76.0 | 65.5 | 71.6 | 69.0 | 56.9 |
| Adaptive | 76.3 | 64.3 | 70.9 | 68.2 | 58.0 |

Table 11: Effect of Gaussian expert count $G$ on model performance

| $G$ | Acc | Edit | F1@{10, 25, 50} | | |
|---|---|---|---|---|---|
| 1 | 76.3 | 65.4 | 72.0 | 69.7 | 58.4 |
| 2 | **76.6** | **66.2** | **72.5** | **70.0** | **59.0** |
| 3 | 76.4 | 65.0 | 71.7 | 68.7 | 58.1 |
| 4 | 75.8 | 64.4 | 71.4 | 68.3 | 57.4 |
| 5 | 75.0 | 65.3 | 70.9 | 67.8 | 57.6 |

Table 12: Effect of window size $w$ on model performance

| $W$ | Acc | Edit | F1@{10, 25, 50} | | |
|---|---|---|---|---|---|
| 5 | 76.3 | 65.1 | 71.1 | 67.9 | 56.7 |
| 10 | **76.6** | **66.2** | **72.5** | **70.0** | **59.0** |
| 20 | 76.0 | 65.3 | 71.8 | 68.6 | 57.0 |
| 40 | 76.2 | 64.5 | 71.4 | 68.9 | 56.8 |
| 80 | 76.2 | 64.3 | 70.3 | 67.4 | 56.3 |

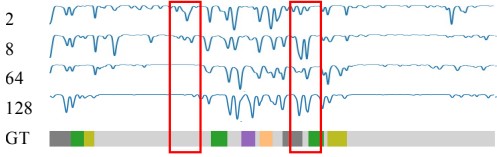

Figure 6: **Curvature Improves Throughout Training.** Early epochs (e.g., 2, 8) exhibit noisy or flat curvature signals, while the converged model (Epoch 64, 128) exhibits distinct low-curvature valleys at action boundaries and high-curvature fluctuations within segments, validating our geometric assumption.

| Epoch | Acc | Edit | F1@10 | F1@25 | F1@50 |
|---|---|---|---|---|---|
| 2 | 74.0 | 73.2 | 78.6 | 75.6 | 64.0 |
| 8 | **74.4** | 73.0 | 78.5 | 74.9 | 63.7 |
| 16 | 74.3 | **74.4** | **79.3** | **76.3** | **65.5** |
| 128 | 74.3 | **74.4** | **79.3** | **76.3** | **65.5** |

Table 13: **Training progression on PKU-MMD (X-Sub).** Curvature-guided features gain discriminability steadily as training proceeds.

Besides, feature quality is critical for curvature estimation. Untrained classification networks produce noisy curvature, so we use staged training: the CGS module activates only after seveal warm-up epochs. We conducted a sensitivity analysis on to find the optimal balance between feature stabilization and joint optimization benefits. Our analysis reveals an optimal warm up epoch for curvature guidance. Starting CGS too early on underdeveloped features is ineffective, while a proper configuration ensures the formation of robust feature clusters before curvature constraints are applied, effectively managing this dependency. As shown in Fig. 6 and Tab. 13 (Appendix C), early curvature is noisy but begins reflecting action boundaries. With continued training, feature compactness improves, further enhancing curvature quality. The gains saturate after a certain number of warm-up rounds.

## D  APPENDIX: REPRODUCIBILITY STATEMENT

We detail the model and training setup in Section 3 and present the dataset, pre-processing, and evaluation protocol in Section 4. All hyper-parameters and computational details are reported in Appendix C. The code is included in the supplementary material.

## E  APPENDIX: LLM USAGE STATEMENT

We utilized a large language model (LLM) to improve the grammar, clarity, and overall readability of this manuscript. The LLM's role was strictly limited to language editing and polishing. All scientific contributions, including the core ideas, methodology, experimental design, data analysis, and conclusions, are the original work of the human authors. The use of the LLM did not alter the scientific content or its interpretation.

