# OpenReview forum: "Curvature-Guided Task Synergy for Skeleton based Temporal Action Segmentation"
_ICLR.cc/2026/Conference — ICLR 2026 Poster_

### Official Review · Reviewer_doPp · 2025-10-28

**Soundness:** 2
**Presentation:** 3
**Contribution:** 2
**Rating:** 4
**Confidence:** 4

**Summary:**

This paper addresses the challenge of conflicting feature requirements in skeleton-based temporal action segmentation (STAS). The key innovation is utilizing the curvature properties of well-learned classification representations on skeleton sequences, with high curvature within action segments and low curvature at transitions serving as geometric priors for boundary detection.

**Strengths:**

This paper proposes a novel curvature-based approach that leverages the geometric properties of feature sequences to foster effective collaboration between classification and localization sub-tasks.

The introduction of a dual-expert weighting mechanism within a Mixture of Experts (MoE) framework enhances the performance of the synergy mechanism by separately capturing semantic representations for classification and fine-grained temporal details for localization.

**Weaknesses:**

There are the following issues with Figure 1: (a) and (b) represent different STAS pipelines, while (c) and (d) do not; (c) and (d) are jumbled together without any spacing to distinguish them; and text in this figure is too small.

Line 231, where, θt ∈ [0, π] quantifies. , should be deleted.

Line 146, Task Decoupling in STAS  . is missing.

There is no analysis of the method's time complexity and runtime. Although a 1.5% improvement is achieved on the PKU-MMD dataset, it would be meaningless if it comes at the cost of increasing the model's runtime.

In temporal action segmentation, are the evaluation metrics Acc, Edit, and F1 reasonable? Why not use evaluation metrics similar to IoU?

**Questions:**

Although the writing in this paper is relatively standardized and the method exhibits some innovations, the task of skeleton-based temporal action segmentation is limited.

This paper does not provide me with meaningful insights, and even if it were accepted, its contribution to the field would be quite limited. I suggest that the authors expand their method for more tasks to enhance its generality.

---

> ### Author Response · Authors · 2025-11-26
> **Response to Reviewer doPp (Part1)**
>
> Thank you very much for your valuable comments and suggestions, which are highly helpful for improving our work. We address your concerns point-by-point below. The additional experiments and discussions will be updated in the revised version.
>
> **W1: Issues with Figure 1 (Layout, Spacing, Text Size)**
> We thank the reviewer for pointing out the layout issues. We have redesigned Figure 1 in the revised manuscript to improve clarity:
> **Structure:** We have adjusted the layout to clearly distinguish the pipeline comparisons ((a) and (b)) from the conceptual illustrations ((c) and (d)).
> **Spacing:** We have added sufficient spacing between (c) and (d) to prevent visual clutter.
> **Readability:** We have significantly increased the font size of all text in the figure to ensure it is easily readable.
>
> **W2: Typo error**
> Line 231 has an extra comma: We have corrected the punctuation in the revised manuscript.
> Line 146 is missing a period: We have added the missing period at the end of the sentence in the revised version.
>
> **W3: model runtimes**
> We appreciate the reviewer's concern regarding the trade-off between performance and efficiency. To address this, we have conducted a comprehensive analysis of Parameters, and runtimes:
>
> **Table: Comparison of Model Efficiency and Performance on PKU-MMD**
> | Method | Acc | Params (M) | FLOPs (G) | Training convergence epochs|Total Training Time (hours) |
> |--------|-------|------------|-----------|----------------------------|----------------------------|
> | Baseline| 69.5 | **1.3** | **11.6**| 300 |18.0 |
> | **Ours** | **71.0** | 1.4| 11.7 | **180** |**10.5** |
>
> Our method introduces negligible additional parameters, as the curvature calculation is parameter-free and the boundary predictor is lightweight. Furthermore, the Gaussian time series experts only need to predict the mean and variance for each expert, requiring a small number of parameters. The geometric prior in our approach not only improves accuracy but also provides more effective training signals, enabling faster convergence without sacrificing model complexity or inference efficiency.
>
> The 1.5% accuracy improvement is achieved with marginal computational cost, proving that our method is both effective and efficient for practical deployment.
>
> **W4: Evaluation Metrics**
> We verify that Acc (Accuracy), Edit (Edit Score), and F1 (Segmental F1 Score) are the standard evaluation protocols for the Temporal Action Segmentation task, widely adopted by the community (e.g., MS-TCN [1]).
>
> Regarding the reviewer's suggestion about IoU, we would like to clarify that our F1 metric is, in fact, an IoU-based metric.
>
> **1. F1 Score is based on IoU:**
> In this task, the F1 score is not the standard frame-wise classification metric. It is the Segmental F1 Score, which specifically measures the quality of predicted segments based on Intersection over Union (IoU).
>
> A predicted segment is considered a True Positive (TP) only if its IoU with the ground-truth segment exceeds a certain threshold (e.g., 10%, 25%, 50%).
> Therefore, by reporting F1@{10, 25, 50}, we are explicitly evaluating the model's performance using IoU constraints.
>
> **2. Why this combination (Acc, Edit, F1) is reasonable?**
> This set of metrics provides a holistic evaluation of different error types, which a single IoU metric might miss:
>
> **Acc (Frame-wise Accuracy):** Measures the global classification correctness.
> **Edit Score:** Measures the segmental continuity and ordering. It heavily penalizes "Over-segmentation" errors (jittery predictions), which is a critical challenge in this field that simple IoU or Acc might not fully reflect.
> **F1 (IoU-based):** Measures the precision of boundary detection and segment overlap.
>
> We adhere to these metrics to ensure a fair and direct comparison with all existing state-of-the-art methods in the literature.
>
> [1] Farha et al., "MS-TCN: Multi-Stage Temporal Convolutional Network for Action Segmentation", CVPR 2019.

---

> ### Author Response · Authors · 2025-11-26
> **Response to Reviewer doPp (Part2)**
>
> **Q1:**  We thank the reviewer for their perspective. We would like to respectfully clarify the significance of the task and the generalizability of our methodological insight, which extends beyond the specific data modality.
>
> **1. Value of Skeleton-based Temporal Action Segmentation:**
> While skeleton-based analysis is a specific sub-field, it is far from limited in its potential impact.
>
> **Privacy & Efficiency:** Unlike RGB videos, skeleton data is privacy-preserving (crucial for healthcare/home monitoring) and computationally lightweight (suitable for edge devices).
> **Robustness:** It remains robust against changing backgrounds and lighting conditions, solving major bottlenecks in RGB-based segmentation.
> Therefore, improving performance in this domain has significant practical value.
>
> **2. Generalizability of the Method (The Core Insight):**
> We argue that our core contribution—"Geometric Curvature as a Supervisor"—is a universal insight that is theoretically applicable to any continuous sequence data, not just skeletons. the proposed Mutual Supervision mechanism (using geometric dynamics to guide semantic consistency) is a general framework that can be readily adapted to other temporal action analysis tasks.
>
> To this end, we conducted additional experiments on the weakly-supervised temporal action localization (WTAL) task, using RGB and optical flow frames as inputs.
>
> We apply the proposed mechanism to a WTAL method [1] and conduct comparisons. Experiments  on THUMOS14 dataset show that the proposed method remains effective.
>
> Table: Validating the proposed task collaboration mechanism on the WTAL task.
> | Method | mAP@IoU 0.3 | mAP@IoU 0.5 | mAP@IoU 0.7 |
> |--------|-------|-------|-------|
> |Boosting[1] | 56.2 | 39.3 | 15.2 |
> | +Ours| **57.1**| **39.7**|**15.5**|
>
> Besides, we also apply the proposed mechanism to RGB video-based temporal action segmentation method [2] and conduct comparisons.  Experiments on Egoprocel dataset show that the proposed method remains effective.
>
> Table: Validating the proposed task collaboration mechanism on the WTAL task.
> | Method | Acc | Edit | F1@10 |F1@25|F1@50|
> |--------|-------|-------|-------|-------|-------|
> |FACT[2] | 55.0|77.2|43.5|45.7|41.3|32.0|
> | +Ours| **56.8**|77.2|**48.2**|**46.0**|**41.9**|**33.3**|
>
> We once again thank you for your valuable comments and suggestions. We hope our responses have addressed your concerns and would be happy to clarify any remaining questions.
>
> [1] Li .et al.  Boosting Weakly-Supervised Temporal Action Localization with Text Information. CVPR 2023.
> [2] Lu .et al, Fact: Frame-action cross-attention temporal modeling for efficient action segmentation. CVPR 2024

---

### Official Review · Reviewer_t1pj · 2025-10-31

**Soundness:** 3
**Presentation:** 2
**Contribution:** 3
**Rating:** 4
**Confidence:** 4

**Summary:**

The paper tackles skeleton-based action understanding by arguing that the geometry of feature sequences over time can be exploited to make classification and temporal localization help each other. The motivation is clear: current methods still show “insufficient cross-task collaboration,” so the authors introduce a curvature-based task synergy mechanism that uses geometric properties of feature trajectories to link both sub-tasks, and they couple it with a dual-expert weighting mechanism in a Mixture-of-Experts setup to adapt features to the task. As the authors themselves note, the approach still struggles with very noisy skeletons and complex multi-person cases, so there is room to improve robustness.

**Strengths:**

Clear motivation.  The paper starts from an identifiable gap, weak collaboration between classification and localization, and address it directly.

Geometric novelty. using curvature of feature sequences as a bridge between sub-tasks is a fresh angle compared to standard temporal smoothing or boundary refinement.

Task-adaptive extraction: the dual-expert / MoE part makes the framework flexible to different task demands instead of using a single shared feature space.

**Weaknesses:**

Major 1. Across the method section, several symbols are introduced without being defined first, or they are not mathematically specified in a precise way. For example:

	•	L176: \alpha_i and \alpha_j are not defined.

	•	L182 – F_s is introduced without stating whether it is the same as the feature sequence in the problem formulation (X) or already an encoded representation.

	•	Eq. (4) and L191 – F_{ST} is used without an earlier, explicit definition. From context, it seems to be the spatio-temporal feature output of the Liformer / GCN stage, possibly the same as F_{\text{gcn}}. Please unify the notation and add a one-sentence definition before Eq. (4).

	•	L223 – “frame-wise classification features.” The text seems to refer to the same tensor that in Fig. 2 is denoted by the decoder classification head Y_{cl}. Please make these two references consistent.

	•	L225–239 – windowed triplets. The text says “three consecutive points,” but the notation is F_{(cls, t-w)}, F_{(cls, t)}, F_{(cls, t+w)}. This is only strictly consecutive when w=1. For w>1, it is unclear: whether intermediate points are also used,  whether triplets are processed independently or aggregated, and how the final task representation is formed from them.

Please provide explicit sampling and aggregation equations.

	•	L262 – F_{st} is mentioned again, but it is not clear whether this is the same as F_s, F_{st}, or F_{ST}. The shape is given as V \times T without channels, which is hard to reconcile with the rest of the model.

⸻

Major 2.
In the ablation results, two entries on PKU-MMD (X-view) — the Edit score and F1@10 — should be bold for the CGS-only configuration. I would like to know why this could be happening, because it would somehow break the idea that “EDD provides the high-quality.”


To make this section coherent:
	1.	Bold those two CGS-only numbers.

	2.	Add a short interpretation, e.g. that CGS might be better aligned with cross-view variability in PKU-MMD than the EDD augmentation in that specific setup.

⸻

Major 3.
	•	Figure 2 seems to have two different flow directions: the central pipeline is read from bottom to top, while the side modules are read from top to bottom. This makes it hard to know where the computation actually starts and where it ends.

	•	Variable names are placed on top of the drawings and are hard to read (small font, low contrast, and sometimes overlapping the boxes).

	•	The figure repeats equations that are already in the text — equations (7) and (8) and L239 — which makes it redundant and not visually explanatory.

	•	The gray vertical line in the first column is not explained . if it is an encoder/decoder split, please state that.

	•	In text (L293) you say the feature is divided into M segments, but the figure shows g_1, g_2, g_3, which suggests a fixed number of Gaussians G=3. Please make the figure consistent with the text.

	•	The final step after the Gaussian generator ends in something like F_{m1}^{ST}, but it is not shown how these features are merged back into the task representation — that is precisely what readers will want to see.

⸻

Minor issues

	1.	L218–220: the statement “intra-segment points must frequently change direction to remain within their class-specific boundary, resulting in high curvature, while inter-segment points exhibit low curvature as they move between class regions” — where is this shown to be true? It would be necessary to include a visualization or some evidence where this can actually be seen.

	2.	L143 – “recent advances” needs citations. If you refer to recent advances, please add representative works.

	3.	Some of the dimension strings in the description are of the form “(V \times K) D \times T \times v” or “1 \times I \times K \, D \times D,” which can be a bit confusing regarding how they are multiplied.

	4.     In the hyperparameter analysis section, the parameter w is not discussed.

	5.	Some sentences are not well expressed or well written. For example:
	•	L216: “…feature space. (This observation can be formally proven: the average curvature of a random walk is inversely proportional to the radius of its bounding hyper-sphere. See Appendix B.) This…”
	•	L448: a sentence should not start with a variable.
	•	The last sentence of L454. Please revise in general.

	6.	Figure 4 – confusing “spins” in parts (b) and (c).
In Fig. 4 (b) and (c) there appear to be many sharp peaks / spins in regions where, according to the action annotation, the action does not change. Please explain why the method produces such high-frequency responses in stable segments.

**Questions:**

1. Notations and definitions. Can you clarify the symbols in the method section (in particular F_s, F_{st}, F_{ST}, F_{\text{gcn}}, and \alpha_i, \alpha_j) and make them consistent with Fig. 2?
2. Figure 2 clarify.
3. Rewrite Ablation on PKU-MMD (X-view): Why does the CGS-only setup outperform the full version for Edit and F1@10?
4. Triplet/window construction: When you say “three consecutive points” but use t-w, t, t+w, what happens for w>1? Are intermediate points used and how are they aggregated into the final task representation?
5. Can you provide a small visualization to support the high-curvature vs low-curvature claim, and explain the extra peaks in Fig. 4 (b)–(c) where the action does not change?

For more details, please see the Weakness section above.

---

> ### Author Response · Authors · 2025-11-26
> **Response to Reviewer t1pj (Part1 Q1, Q4 & Major1)**
>
> Thank you for your guidance and suggestions. They are very helpful for improving our work. We address your questions point by point below. We address each point in detail below. The additional experiments and discussions will be updated in the revised version.
>
> **Q1, Q4 & Major1:**  We apologize for the confusion caused by the notation inconsistencies. We have thoroughly revised the Method section to ensure mathematical rigor. Below is the clarification for the symbols and the updated definitions:
>
> **Symbol Clarifications**
> **1. $\alpha_i, \alpha_j$   (Line 176)**:
> These represent the indices of joints within the skeleton graph. The term d($\alpha_i, \alpha_j$ ) denotes the shortest geodesic distance between joint i and joint j in the skeletal topology. Since the foundation framework is built upon existing methods, we have simplified **Section 3.2** as suggested by reviewer hoUk. The symbol  $\alpha_i, \alpha_j$ no longer appears in the revised version.
>
>
> **2. F_s (Line 182)**:
> This represents the original input skeleton sequence (previously denoted as $X$ in the problem definition). For consistency throughout this document, we will uniformly represent the original input skeleton sequence as $F\_s$ with a shape of ($D\_{in} \times T \times  V$) (where $D\_{in}$ = 3 for 3D coordinates).
>
>
> **3. F_{ST}  vs. F_{gcn}  (Line 191, Eq. 4)**:
> They are distinct. **F_{gcn}** is the output of the Spatial Modeling module (MS-GCN), capturing purely spatial dependencies. F_{ST} is the global spatio-temporal feature obtained after integrating **F_{gcn}** with spatio-temporal embeddings via the spatio-temporal attention fusion. The flow is:
>
> **F_s** (Input) -> MS-GCN ->**F_{gcn}** (Spatial Encoder) -> Spatio-Temporal Attention Fusion -> **F_{ST}** (Spatio-Temporal Encoder)
>
> We have added the explicit definition of **F_{ST}** before Eq. (4) as suggested.
>
> **4. Clarification on F_{ST}, F_{cls}, and Y_{cl}:**
> **1) F_{ST}** (frame-wise Feature): **F_{ST}**(specifically **F_{ST}^L**) is the generic output of the encoder backbone with shape ($D \times T$).
>
> **2) F_{cls}** (Task-Specific Feature): The term "frame-wise classification features" (Line 223) refers to **F_{cls}**. In our EDD (Expert Decoupling) module, the global feature **F_{ST}^L** is decoupled into two task-specific representations: **F_{cls}** for the classification expert and **F_{brb}** for the localization expert. Thus, **F_{cls}** is distinct from **F_{brb}**.  Details of the EDD module can be found in Section 3.4.
>
> **3) Y_{cl}** (Prediction Logits): **Y_{cl}** in Figure 2 represents the final probability logits ($C\times T$, $C$ denotes the number of action categories) produced by the classification head. The relationship is **Y_{cl}**= Head_cls(**F_{cls}**). We have clarified that **F_{cls}** is the decoupled latent feature, while **Y_{cl}** is the predictive output in our revised version.
>
> **5.  Q4 (Windowed Triplets Construction)**
> Regarding the curvature calculation with window size $w>1$, we employ a robust two-stage strategy to capture multi-scale geometric properties while suppressing noise:
>
> **Firstly**, for a given center frame *t*, we construct a triplet using indices (t-w, t, t+w) (that is, even if w>1, we only use the features from the t-wth frame, the tth frame, and the t+wth frame to calculate the curvature of the tth frame's feature. **This is equivalent to downsampling the features in the time dimension before calculating the curvature**.  By calculating the curvature of the triangle formed by these stride feature points, the model can capture the macro-level geometric information of the trajectory, thereby making it robust to local temporal jitter.
>
> Furthermore, after calculating the initial curvature sequence, we apply a **temporal moving average filter** to the final curvature values (using the same window size w, (i.e., **averaging the feature curvatures of the 2w-1 frames within the window to replace the curvature value at the t-th frame**). This post-processing step further suppresses high-frequency noise and ensures a stable guidance signal.
>
> **6.Correction on L262 (F_{st} shape)**
> You are right. I apologize for the typographical error. The tensor **F_{st}** is the spatio-temporal feature **F_{ST}**, which has a shape of $D \times T \times V$ (channel-first representation). We have corrected this error in the revised version.

---

> ### Author Response · Authors · 2025-11-26
> **Response to Reviewer t1pj (Part2 Q2 & Major3, Q3 & Major2)**
>
> **Q2 & Major3:** We are grateful for the reviewer's detailed feedback on Figure 2. We acknowledge that the original visualization was cluttered and logically inconsistent. We have completely redrawn Figure 2 to improve clarity and flow in the revised version. Specifically:
>
> **Flow Direction:** We have unified the data flow direction. All modules (including the central pipeline and side experts) now follow a consistent bottom-to-top flow to match the logical progression of the network.
> **Readability:** We have increased the font size and repositioned variable names to avoid overlapping with graphical elements.
> **Redundancy:** Explicit equations (e.g., Eq. 7, 8) have been removed from the figure to reduce visual clutter.
> **Structure:** We have explicitly labeled the gray vertical line as the boundary between the "Spatio Encoder" and "Temporal Eecoder" to clarify the architecture split.
> **Consistency:** The visualization of the Gaussian segments has been updated. Instead of implying a fixed number of three, we now use notations like F^{(m)}, g^{(m)} to represent a generic number of segments or gaussian filters, ensuring consistency with the text description.
> **Final Step of EDD:** The spatio-temporal representation F_{ST} is split along the temporal dimension into $M$ segment features $F^{(m)}$. The final outputs $\tilde{F}^{(m)}$ from the Gaussian generator are then concatenated along the temporal dimension back into the task representation (e.g., F_{cls}). We have added a brief explanation in the figure to clarify this merging step.
>
> We believe the new Figure 2 provides a much clearer and more accurate representation of our method.
>
> **Q3 & Major2:** We thank the reviewer for this insightful observation. We have revised the paper to bold the CGS-only entries (Edit and F1@10) and added a concise interpretation in the "Synergy of the Full Model" section.
>
> The phenomenon on PKU-MMD (X-view) reveals a trade-off between prediction continuity and localization precision under drastic view shifts:CGS-only operates on shared features, which are spatially smoother. In the difficult X-view setting, this smoothness aids in generating continuous predictions (favoring Edit scores) and achieving rough overlaps (favoring loose F1@10).
> Full Model (with EDD) utilizes specialized expert features that are significantly sharper. While this sharpness makes predictions slightly more sensitive to view-induced noise (marginally affecting Edit scores), it is crucial for precise boundary delineation. This is confirmed by the Full Model's superior performance on strict metrics (F1@25, F1@50) and Accuracy.

---

> ### Author Response · Authors · 2025-11-26
> **Response to Reviewer t1pj (Part3  Minor1,Minor2,Minor3,Minor4, and Minor5)**
>
> **Minor 1:**  We appreciate the reviewer for raising this question.
> **Clarification & Evidence:**
> Our method is built on the simplified manifold assumption derived in **Appendix B (Page 15-17 in revised version)**, which mathematically proves that features exhibit high variance (high curvature) within the class manifold due to intra-class diversity, but follow a smoother geodesic path (low curvature) when transitioning between distinct class manifolds.
>
> To demonstrate this, we refer to **Figure 6** in the revised manuscript **(Appendix C, Page 18)**, which visualizes the evolution of the curvature signal over time. Specifically,  For the curvature trajectory of the fully trained classificaltion model (e.g., Epoch 128 in Figure 6), the visualization clearly shows the predicted behavior:
>
> **Boundaries as Valleys:** The curvature signal drops significantly, forming distinct "valleys" (low curvature) that align precisely with the ground truth action boundaries (vertical dashed lines).
> **Intra-segment High Curvature:** Conversely, within action segments, the signal maintains higher values due to the complex variance of intra-class features. This empirical evidence directly supports the statement in Line 218–220 and aligns with our theoretical proof provided in the Appendix.
>
> **Minor 2:**  We apologize for the oversight in the Related Work section. Although we discussed state-of-the-art methods like DeST and LaSA in our Introduction and Method sections, we failed to include a dedicated discussion of them in Section 2. In the revision, we will restructure Section 2 to include a comprehensive review of these recent advancements such as [1],[2] and [3].
>
> [1] Yunheng Li, Zhongyu Li, Shanghua Gao, Qilong Wang, Qibin Hou, and Ming-Ming Cheng. A decoupled spatio-temporal framework for skeleton-based action segmentation. arXiv preprint arXiv:2312.05830, 2023c.
> [2] Di Yang, Yaohui Li, et al. Latent action composition for skeleton-based action segmentation. In Proceedings of the IEEE/CVF International Conference on Computer Vision (ICCV), pp. 10000–10010, 2023.
> [3] Haoyu Ji, Bowen Chen, Xinglong Xu, Weihong Ren, Zhiyong Wang, and Honghai Liu. Language-assisted skeleton action understanding for skeleton-based temporal action segmentation. In European Conference on Computer Vision, pp. 400–417. Springer, 2024.
>
> **Minor 3:**  We apologize for the confusion regarding the dimension notations. We originally adopted specific formats (e.g., "$1 \times 1 \times KD$", "$V \times KV$") directly from the baseline paper DeST[1] and LaSA[2] to maintain consistency with their variable definitions, where K represents the number of hops and $D$ represents channels. However, we agree with the reviewer that this mixed notation can be ambiguous regarding scalar multiplication versus tensor dimensions.
>
> We have revised these descriptions throughout the paper to use standard mathematical notation. Specifically: We have added explicit parentheses and multiplication dots to clarify combined dimensions (e.g., changing "$KD$" to "($K \cdot D$)"). This strictly distinguishes dimension size calculations from tensor axes. We have standardized the description of convolution kernels to clearly separate kernel size, input channels, and output channels. These changes ensure the mathematical definitions are unambiguous while retaining the necessary logical connection to the baseline method.
>
> **Minor 4:** We have added a sensitivity analysis for the hyperparameter w (window size) in Appendix C. We also provided a brief analysis in the main text **(Section 4.2, Page 9, line 473)**: Hyper-parameter $w$ controls the temporal receptive field. We found $w$=10 optimal, balancing noise suppression and boundary sensitivity. Smaller windows ($w$=5) capture insufficient context, while larger ones ($w$>=40) over-smooth the directional changes that define action boundaries.}
>
> **Minor 5:** We sincerely apologize for the writing errors and stylistic inconsistencies in the original manuscript. We appreciate the reviewer's detailed attention to these issues. We have carefully proofread the entire paper and specifically revised the sentences mentioned as follows:
> "Line 216:"...confine action sequence trajectories within compact, class-specific regions. As formally derived in Appendix B, this geometric constraint implies that the average curvature of a random walk is inversely proportional to the radius of its bounding hyper-sphere."
> "Line 448:  "The hyperparameter \lambda balances the contribution of our Curvature Guidance
> "Line 454:  "...Detailed hyperparameter analysis is provided in Appendix C."

---

> ### Author Response · Authors · 2025-11-26
> **Response to Reviewer t1pj (Part4 Q5 & Minor6)**
>
> **Q5 & Minor 6:** We thank the reviewer for this careful examination.
>
> The high-frequency fluctuations ("Spins") observed in the stable segments of Figure 4 are expected and attributable to the "complexities of the real world." semantic stability (the same action label) does not imply geometric stillness. Real-world actions, especially the "background" category shown in the long segment of Figure 4(b), encompass a wide variety of unconstrained movements. The observed "Spins" indicates that the curvature metric is sensitive, capturing these subtle geometric variations and input noise within the action sequence, rather than over-smoothing the signal.
>
> Although the similarity between adjacent frames fluctuates (causing curvature spins), these frames remain significantly closer to their ground-truth class center than to any other class center.As long as the feature trajectory, however jittery, stays within the "territory" of the correct class cluster and does not cross the decision boundary towards another class, the classification remains stable. Besides, our mutual supervision mechanism between curve and the output of boundary prediction head ensures the system correctly distinguishes these intra-class spins from true boundaries.
>
> We once again thank you for your valuable comments and suggestions. We hope our responses have addressed your concerns and would be happy to clarify any remaining questions.

---

### Official Review · Reviewer_hoUk · 2025-10-31

**Soundness:** 3
**Presentation:** 3
**Contribution:** 3
**Rating:** 6
**Confidence:** 4

**Summary:**

This paper addresses a core challenge in Skeleton-based Temporal Action Segmentation (STAS): the conflicting feature requirements for its two main sub-tasks, action classification (which needs temporal invariance) and boundary localization (which needs temporal sensitivity). Existing methods typically decouple these tasks, which prevents beneficial cross-task collaboration and creates "information silos". The paper proposes CurvSeg, a novel approach that synergizes these tasks using a geometric curvature guidance mechanism. The key insight is that in a well-learned classification feature space, the trajectory of skeleton frame features exhibits high curvature within an action segment (to stay within its class cluster) but low curvature at transitions when moving between clusters.

**Strengths:**

1.The paper introduces a novel curvature-based task synergy mechanism (CGS) that effectively exploits the geometric properties of feature sequences. This mechanism establishes a self-reinforcing loop where improved boundary detection and more discriminative classification features mutually enhance one another.

2. The method is validated through comprehensive experiments on multiple benchmark datasets (PKU-MMD, LARa, MCFS-22, and MCFS-130) , where it achieves superior, state-of-the-art results. The most significant gains are seen in segmental F1 scores, directly validating the method's ability to enhance temporal boundary precision.

3. Thorough ablation studies demonstrate the efficacy of each core component. The studies show that both the Expert-Driven Decoupling (EDD) and the Curvature-Guided Synergy (CGS)  independently improve performance. When combined, the full model achieves a synergistic effect, with performance gains surpassing the sum of the individual modules.

4. The paper demonstrates that curvature is a more robust proxy for action boundaries than traditional distance metrics like Euclidean or Cosine. Curvature is sensitive to changes in the direction of the feature trajectory, making it better at detecting both gradual and abrupt action transitions.

**Weaknesses:**

1. The related works in Skeleton-based Temporal Action Segmentation are not fully discussed. Only two works in 2020 are discussed. More recent works shuold be incorporated.
2. As reflected by Eq.9, the information of the classification head and localization head just interact once, not in a self-reinforcing loop as the authors describe.
3. The description of the foundation framework in sec.3.2 owns too much space. As the baseline and foundational model, it should be compactly introduced.
4. The key innovation is directly borrowed from the previous work (Shinet al., 2024), and adopted for STAS with a simple transfer.
5. Although the paper asserts that high curvature corresponds to intra-segment motion and low curvature to boundaries, the theoretical link is only qualitatively motivated and relies on assumptions (e.g., class clusters as hyperspheres). The “Appendix B proof” simplifies dynamics to random walks within spheres, which is too idealized for real, noisy skeleton trajectories.

**Questions:**

See above

---

> ### Author Response · Authors · 2025-11-26
> **Response to Reviewer hoUk (Part1 W1,W2,W3 and W4)**
>
> Thank you very much for your positive recognition of our work. Your suggestions and questions are highly valuable and greatly help us further improve the paper. We address each point in detail below. The additional experiments and discussions will be updated in the camera-ready version.
>
> **W1:** We apologize for the oversight in the Related Work section. Although we discussed state-of-the-art methods like DeST and LaSA in our Introduction and Method sections, we failed to include a dedicated discussion of them in Section 2. In the revision, we will restructure Section 2 to include a comprehensive review of these recent advancements such as [1],[2] and [3].
>
> [1] Yunheng Li, Zhongyu Li, Shanghua Gao, Qilong Wang, Qibin Hou, and Ming-Ming Cheng. A decoupled spatio-temporal framework for skeleton-based action segmentation. arXiv preprint arXiv:2312.05830, 2023c.
> [2] Di Yang, Yaohui Li, et al. Latent action composition for skeleton-based action segmentation. In Proceedings of the IEEE/CVF International Conference on Computer Vision (ICCV), pp. 10000–10010, 2023.
> [3] Haoyu Ji, Bowen Chen, Xinglong Xu, Weihong Ren, Zhiyong Wang, and Honghai Liu. Language-assisted skeleton action understanding for skeleton-based temporal action segmentation. In European Conference on Computer Vision, pp. 400–417. Springer, 2024.
>
> **W2:** We apologize for the confusion caused by the term "self-reinforcing loop." We did not mean to imply a recursive architecture (like an RNN) or an iterative inference procedure where heads exchange information multiple times within a single forward pass. The "Loop" refers to the Training Dynamics.
>
> Instead, we refer to the synergistic evolution of the two tasks throughout the training process (over epochs and iterations):
>
> **Cls → Loc:** The classification features provide curvature cues to guide the localization head (via the CGS module).
> **Loc → Cls:** The localization head predicts boundaries, which are then used to supervise the classification features (penalizing low curvature inside predicted segments).
> **The Cycle:** In subsequent iterations, the improved classification features yield cleaner curvature signals, which further refine localization, which in turn imposes better constraints on the feature space.
>
> While the interaction in **Eq. 7** (**page 5, line 248 of revised version**) happens once per forward pass, the effect accumulates over thousands of gradient steps. The "loop" is closed via the gradient updates: the improvement in one head creates a better training signal for the other in the next step, where components improve iteratively despite having fixed connections.
>
> **Revision:**
> We have rephrased the relevant section to avoid the term "loop" and instead use "synergistic training strategy" to clearly our method.
>
>
> **W3:** We fully agree with the reviewer. Since the backbone follows established architectures (DeST, LaSA), extensive detailing is unnecessary. In the **revision**, we have significantly condensed Section 3.2. Specifically, we  streamlined the details of  the baseline backbone architectures, focusing only on the essential input/output dimensions and data flow.
>
> **W4:** We acknowledge the inspiration from Shin et al. regarding the geometric insight that curvature can signal boundaries.    However, our work fundamentally transcends a simple transfer by repurposing this observation as a synergistic bridge to address the critical limitation of insufficient cross-task collaboration in existing STAS methods.   While Shin et al. validated curvature as a passive boundary indicator, we engineer a closed-loop mechanism where this geometric property actively connects classification and localization.  Specifically, we establish a bi-directional dependency: classification features guide boundary detection via curvature, while localization predictions reciprocally supervise the feature space to enforce this geometric structure.  This transforms a static observation into a dynamic optimization strategy, offering a systematic solution to multi-task alignment that is absent in prior temporal action segmentation works.

---

> ### Author Response · Authors · 2025-11-26
> **Response to Reviewer hoUk (Part2 W5)**
>
> **W5:**  We thank the reviewer for pointing out the idealization in our theoretical model. We clarify the link between the theory and real-world data from three perspectives:
> **1. Role of the Theoretical Model (Justifying Assumptions):**
> The derivation is intended as a theoretical abstraction to provide geometric intuition, capturing the essential core of the representation learning process:
>
> **1) Hyperspherical Assumption:** While real clusters are not perfect spheres, the training objective (Cross-Entropy and Contrastive Learning in baselines) explicitly minimizes intra-class variance. This optimization naturally shapes features into compact, quasi-spherical clusters [Wang et al., 2022], making the sphere assumption a valid first-order approximation.
>
> **2) Random Walk vs. Structured Motion:** We used the "random walk" model to prove a lower bound: even under stochastic motion, confinement induces curvature. Real learned trajectories are more structured (aiming to stay within the cluster center), which effectively reinforces our conclusion: the purposeful avoidance of boundaries in real data yields even higher curvature signals than a random walk.
>
> [1] Wang Y, Zhang Q, Wang Y, et al. Chaos is a Ladder: A New Theoretical Understanding of Contrastive Learning via Augmentation Overlap[C]//International Conference on Learning Representations.
>
> **2. Addressing "Noisy Skeleton Trajectories" (The Filtering Effect):**
> The reviewer correctly notes that raw skeleton data is noisy. However, it is crucial to clarify that we compute curvature in the high-level Semantic Feature Space, not on raw coordinates. The backbone network acts as a non-linear filter. Through layers of aggregation, high-frequency jitter from the raw skeleton input is suppressed. Consequently, the remaining trajectory reflects semantic evolution, which is significantly smoother and fits the "continuous manifold" assumption much better than raw data.
>
> **3. Verification (Qualitative & Quantitative):**
> **Quantitative:** The most rigorous validation is the segmentation performance. The significant F1 score improvements (Tables 1 & 2) demonstrate that the curvature-based prior effectively captures boundary signals in real-world data, validating the theory's effectiveness.
>
> **Qualitative:** Figure 4(a) shows a representative case where the geometric trend clearly aligns with our theory. Figure 4(b) illustrates a challenging scenario with the "noisy" dynamics noted by the reviewer. Crucially, valid boundary signals are still partially present. Through mutual supervision, predicted boundaries suppress feature noise by enforcing compactness, while curvature provides geometric guidance, jointly ensuring robustness against such variance.
>
> We once again thank you for your valuable comments and suggestions. We hope our responses have addressed your concerns and would be happy to clarify any remaining questions.

---

### Official Review · Reviewer_7JNX · 2025-11-14

**Soundness:** 3
**Presentation:** 3
**Contribution:** 3
**Rating:** 6
**Confidence:** 2

**Summary:**

This paper proposes CurvSeg, a novel framework for skeleton-based temporal action segmentation (STAS) that addresses the long-standing tension between action classification and boundary localization. The key insight is geometric: well-separated classification features naturally induce high curvature within action segments and low curvature at transitions — forming a "valley" that serves as a strong prior for boundary detection.
To exploit this, the authors introduce two core components:
Curvature-Guided Synergy (CGS): A bidirectional consistency mechanism where classification feature curvature guides boundary prediction, while boundary supervision regularizes classification features to enhance cluster compactness.
Expert-Driven Decoupling (EDD): A Mixture-of-Experts module with task-specific experts that refine shared encoder outputs into adaptive representations for classification and localization.
Extensive experiments on four benchmarks (PKU-MMD, LARa, MCFS-22/130) show consistent improvements over state-of-the-art methods, particularly in segmental F1 scores, validating the effectiveness of curvature-guided collaboration.
The work makes a compelling case for geometric priors in structured prediction tasks, offering both conceptual novelty and practical gains.

**Strengths:**

Originality

Innovative use of representation geometry: Leveraging trajectory curvature as a cross-task signal is conceptually fresh and theoretically grounded (Appendix B). This moves beyond typical attention or fusion mechanisms.
Bidirectional synergy design: Unlike prior decoupled frameworks that treat tasks independently, CurvSeg establishes a mutual reinforcement loop, which is rare in STAS literature.
Task-adaptive MoE without parameter explosion: The Gaussian expert routing is lightweight yet effective, enabling dynamic feature specialization without full dual-path architectures.

 Quality

Rigorous experimental evaluation: Results across four datasets, including ablation studies and comparisons with strong baselines (DeST, LaSA), demonstrate robust performance gains.
Well-designed ablations: Tables 3–6 clearly isolate contributions of CGS and EDD, showing their individual and synergistic effects.
Qualitative visualization: Figure 4 effectively illustrates improved boundary precision and reduced over-segmentation.

 Clarity

The paper is well-written and logically structured, with intuitive figures (Fig. 1–2) explaining the core ideas.
Equations are clearly presented, and Algorithm 1/2 provide sufficient implementation details.
Appendices offer valuable theoretical justification and hyperparameter analysis.

Significance

Addresses a fundamental limitation in STAS: insufficient cross-task collaboration despite semantic interdependence.
Demonstrates that geometric structure in learned representations can be exploited for downstream tasks — an idea potentially applicable to other time-series problems (e.g., speech segmentation, medical signal analysis).
Offers a new paradigm: using internal model dynamics (curvature) as supervisory signals, reducing reliance on external priors.

**Weaknesses:**

1) Limited discussion on failure cases
While the method performs well overall, there is no analysis of when or why curvature fails as a boundary proxy. For example:
In gradual transitions (e.g., slow hand movement), curvature may not form clear valleys.
Noisy skeleton data might amplify spurious curvature peaks.
A brief error analysis (e.g., per-action performance drop) would strengthen the claims.

2) Assumption of uniform segment partitioning
The EDD module divides videos into fixed-length segments (e.g., M=64), regardless of actual action duration. This could misalign temporal patterns for very short or long actions. Some discussion on adaptivity (e.g., content-aware segmentation) would improve robustness.

3) Dependency on classification quality
The CGS module assumes that classification features already form compact clusters. If initial clustering is poor (e.g., due to ambiguous actions), curvature may not emerge reliably. The paper lacks sensitivity analysis under weak classification regimes.

 4) Reproducibility concerns
Although code will be released, some implementation details are missing:
How exactly are classification features $F_{cls}$ extracted? From the encoder output or after the classification head?
Is the curvature computed per-joint or globally?
These should be clarified in the final version.

**Questions:**

Q1: In Section 3.3, you mention that low-curvature regions correspond to boundaries. But in Fig. 4(a), we see high curvature at boundaries. Could you clarify this apparent contradiction? Is it possible that both high and low curvature can indicate transitions depending on context?
This could change my understanding of whether curvature acts as a direct boundary detector or only an indirect regularizer.

Q2: You show in Table 5 that curvature outperforms Euclidean/Cosine distance metrics. Have you considered comparing against learned boundary detectors (e.g., gradient-based saliency maps)? Does curvature still dominate in such comparisons?

Q3: What happens if you apply the curvature signal only during training but remove it at inference? Would performance drop significantly? This would help quantify how much of the gain comes from architectural synergy vs. test-time guidance.

---

> ### Author Response · Authors · 2025-11-26
> **Response to Reviewer  7JNX (Part1 W1,W2,W3)**
>
> Thank you very much for your valuable comments and suggestions, which are highly helpful for improving our work. We address your concerns point-by-point below. The additional experiments and discussions will be updated in the revised version.
>
> **W1:** We have added a detailed limitations analysis in **Section 4.5.2 Error Analysis and Limitations**(**revised paper, page 9, line 477**), and supplemented it with corresponding **qualitative** and **quantitative** analyses (**revised paper, top of page 10**).
>
> **Quantitative Analysis :**
> To quantify the impact of motion dynamics, we analyzed the MCFS dataset by mapping specific skating actions to dynamic categories:
> **1.High-Dynamic (Distinct):** Jumps & Spins (e.g., 3Axel, SitSpin). These actions involve rapid velocity changes and distinct poses, creating sharp geometric boundaries.
> **2.Low-Dynamic (Gradual):** Steps & Transitions (e.g., ChoreoSequence, StepSequence). These involve continuous, flowing body movements with smooth transitions.
> **Table: Performance Comparison by Motion Dynamics on MCSF**
> | Category | Metric | Base | Ours |
> |----------|--------|------|------|
> | **High-Dynamic** | F1@10 | 71.18 | **72.86** |
> | | Edit | 58.44 | **61.61** |
> | **Low-Dynamic** | F1@10 | 49.62 | **52.86** |
> | | Edit | 21.81 | **27.83** |
>
> As hypothesized, we observe a significant performance drop in Low-Dynamic actions **52.86%** compared to High-Dynamic ones **72.86%**. This confirms that "gradual transitions" produce flatter curvature profiles, making boundary detection physically harder.  Surprisingly, we gain **+6.0%** Edit Score on Low-Dynamic actions, compared to only **+3.2%** on High-Dynamic ones. That's because in high-dynamic actions, baseline features suffice. In gradual transitions, ambiguous features cause it to fail. Our curvature prior then guides the model: even weak signals provide structural guidance, enabling more precise boundary localization than the baseline.
>
> **Qualitative Analysis​:**
>  As shown in Fig. 4  in revised version, in gradual transitions (e.g., "Step Sequence"), continuous motion creates smooth curvature without clear valleys, making boundaries ambiguous. Conversely, skeleton noise can introduce false peaks. While challenging, our method proves robust, maintaining reliable performance where baselines fail.
>
> **W2:** Thank you for noting the rigidity of fixed-length segments. We have added additional adaptive segment number sensitivity analysis in **Appendix C(revised paper, bottom of page 17)**.
> **Table: Effect of segment count $M$ on model performance**
> | M    | Acc  | Edit | F1@10 | F1@25 | F1@50 |
> |--------|------|------|-------|-------|-------|
> | 8      | 76.3 | 64.9 | 71.5  | 68.3  | 56.7  |
> | 16     | 75.5 | 64.6 | 71.5  | 68.4  | 56.8  |
> | 32     | 76.4 | 65.9 | 72.2  | 69.7  | 58.3  |
> | 64     | **76.6** | 66.2 | **72.5** | 70.0 | **59.0** |
> | 128    | 76.0 | 65.5 | 71.6  | 69.0  | 56.9  |
> | adapt  | 76.1 | **66.5** | 72.4 | **70.1** | 58.7 |
>
> **Analysis:**
> Performance remains stable across segment counts (M=32,64,128), indicating robustness to boundary placement. This stems from **Gaussian Experts' micro-level adaptivity**. While global partitioning provides coarse structure, these experts act as dynamic filters: they focus on specific frames for short actions (small $\sigma$) and shift centers ($\mu$) to "stitch" across boundaries. As continuous soft masks, they capture patterns flexibly within fixed windows.
>
> Thus, internal adaptivity compensates for rigid boundaries. Given adaptive partitioning's computational cost with minimal gains, we select fixed M=64 for optimal efficiency-accuracy balance.
>
> **W3:** We agree that feature quality is critical for curvature estimation. Untrained classification networks produce noisy curvature, so we use staged training: the CGS module activates only after $E\_w$&nbsp;&nbsp;warm-up epochs. We conducted a sensitivity analysis on $E\_w$&nbsp;&nbsp; to find the optimal balance between feature stabilization and joint optimization benefits.
>
> Our analysis reveals an optimal warm up epoch $E\_w$ for curvature guidance. Starting CGS too early on underdeveloped features is ineffective, while a proper configuration ensures the formation of robust feature clusters before curvature constraints are applied, effectively managing this dependency.
>
> As shown in Fig. 6 and Tab. 13 (Appendix C), early curvature is noisy but begins reflecting action boundaries. With continued training, feature compactness improves, further enhancing curvature quality. The gains saturate after a certain number of warm-up rounds.
>
> **Table: Effect of warm up epoch on model performance on PKU-MMD(x-sub)**
> | $E\_w$   | Acc  | Edit | F1@10 | F1@25 | F1@50 |
> |--------|------|------|-------|-------|-------|
> | 2      | 74.0 |73.2 | 78.6 | 75.6  | 64.0  |
> | 8      | 74.4 | 73.0 | 78.5  | 75.3 | 63.7 |
> | 16    | 74.3 | 74.3 | 79.3  | 76.3 | 65.5 |
> | 128  | 74.3 | 74.3 | 79.3  | 76.3 | 65.5 |

---

> ### Author Response · Authors · 2025-11-26
> **Response to Reviewer 7JNX (Part2 W4,Q1,Q2,Q3)**
>
> **W4:** We thank the reviewer for pointing out these missing details. We will clarify them in the revision:
>
> **Feature Extraction:** The classification features F_{cls} are extracted directly from the output of the spatio-temporal encoder (specifically, the output of the classification expert in our MoE structure), before entering the final classification head.
> **Curvature Scope:** The curvature is computed globally per frame, not per joint. The joint dimension V is aggregated via global average pooling after the spatial modeling stage, resulting in a frame-wise feature vector
> for curvature calculation.
>
> **Q1:**  We appreciate the reviewer's careful examination of Fig. 4(a).
> **1. Clarifying the Geometric Assumption:**
> Our work operates on the specific hypothesis inspired by [Shin et al., 2024] that within a well-clustered action segment, the feature trajectory is confined and must continuously "turn" to stay within the cluster, resulting in high curvature. Conversely, during a transition (boundary), the trajectory traverses the empty space between clusters via a relatively straight path (geodesic), resulting in low curvature.
>
> **2. Addressing the Visual Confusion in Fig. 4(a):**
> When the reviewer notes "high curvature at boundaries," we suspect two reasons:
>
> **1)Visual Sharpness:** The Y-axis of the first row in Fig. 4(a) plots the absolute magnitude of the feature curvature. Consequently, the "valleys" at boundaries—while appearing as visually sharp spikes due to the rapid drop and rise of the signal—actually correspond to regions where the curvature magnitude drops to a local minimum. The "sharpness" the reviewer observed reflects the distinctness of the boundary signal, but the low numerical value at the valley's bottom is what mathematically indicates the transition.
>
> **2) Alignment with GT:** There might be slight temporal offsets between the predicted valleys (Ours) and the exact GT boundaries. Since boundary detection allows for some tolerance (evaluated via F1@IoU), the lowest point of the curvature valley aligns with our predicted boundary, even if the GT boundary falls slightly to the side where the curve is rising.
>
> There is no contradiction in the mechanism itself. The "low curvature = boundary" premise is consistent with our manifold assumption. The "high visual activity" (spikes) at boundaries in the plot actually represents the target "low metric values" we seek. We have revised the caption of Figure 4 to explicitly state: "The Y-axis represents curvature magnitude  to clarify this confusion.
>
> **Q2:** We appreciate this suggestion. Following your advice, we incorporated a Gradient-based Saliency baseline into our comparison (**Tab. 4 in the revision, page 8**). This baseline uses feature gradients to generate the guidance signal for the boundary branch. As shown in the revised table below, our curvature-based method also outperforms gradient-based saliency methods.
>
> **Table: CGS ablation studies on LARa dataset**
> | Metric | Acc | Edit | F1@10 | F1@25 | F1@50 |
> |--------|-----|------|-------|-------|-------|
> | Base | 75.3 | 65.7 | 71.6 | 69.0 | 57.9 |
> | Euclid | 76.0 | 65.1 | 71.7 | 68.9 | 57.8 |
> | Cosine | 75.2 | 65.2 | 71.5 | 68.6 | 57.0 |
> | Grad. Saliency | 74.4 | 64.3 | 70.7 | 67.9 | 57.1 |
> | **Our Curv** | **76.2** | **65.9** | **72.1** | **69.4** | **58.7** |
>
> The performance gap stems from a spatial misalignment of the signals: Gradient saliency highlights the most discriminative frames (e.g., the apex of a jump) to maximize classification confidence. These peaks typically occur in the middle of actions and are therefore not sensitive to action boundaries. In contrast, curvature specifically measures the change in direction of feature trajectories. It focuses on the evolution of states, making it a more precise proxy for temporal boundaries.
>
> **Q3:**  We clarify that in our framework, Curvature Guidance serves strictly as a training-time supervision signal. During inference, the curvature module is deactivated, and predictions rely solely on the trained boundary branch. Thus, our reported results already reflect the "removal" of curvature at inference.
> To quantify the value of this geometric supervision, we evaluated the raw inverted curvature directly as a boundary predictor in **Section 4.5.1 Tab.5 of revised version**. This curvature achieves competitive performance. This strongly confirms that low-curvature points in the feature trajectory are a highly effective proxy for temporal boundaries, validating the foundational principle of our CGS module.
>
> **Table. Refine with different boundary prediction.**
> | Metric | Acc | Edit | F1@10 | F1@25 | F1@50 |
> |--------|-----|------|-------|-------|-------|
> | Boundary Prediction | 76.6| 66.2 |72.5 |70.0 |59.0|
> | Curvature | 75.4 |64.6 |72.3| 69.4 |55.1|
>
> We once again thank you for your valuable comments and suggestions. We hope our responses have addressed your concerns and would be happy to clarify any remaining questions.

---

### Author Response · Authors · 2025-11-26
**Global Response**

We sincerely thank all reviewers for their constructive feedback. We are encouraged that the reviewers unanimously recognized our work as **"conceptually fresh" (Reviewer 7JNX)** and possessing **"geometric novelty" (Reviewer t1pj)**, validated our **"novel curvature-based task synergy" (Reviewer hoUk, Reviewer doPp)**, and appreciated the **"rigorous experimental evaluation" (Reviewer 7JNX)** that demonstrates **"superior results" (Reviewer hoUk)** on multiple benchmarks.

**Here is a categorized summary of key issues raised by the reviewers:**

**I. Clarification of Mechanisms and Core Principles**
**R-7JNX (W1, W3; Q1):** Limitation analysis of curvature for smooth transitions/noisy clusters, and clarification of high boundary curvature in Fig. 4(a).
**R-hoUk (W2, W4, W5):** Key questions addressed include the "self-enhancement" property (Eq. 9), idealized proof assumptions (Appendix B), and distinctions from Shin et al. (2024).
**R-t1pj (Minor 1, 6; Q5):** Asks why the "high-frequency spin/spikes" appearing within stable action segments in Fig 4; and requests visual evidence supporting the claim that "low curvature corresponds to boundaries."

**Our summary of responses:**
We clarified the "synergistic training" nature of our mechanism and differentiated our active bi-directional optimization from Shin et al., explaining how the classifier effectively filters intra-class "spins" to ensure robustness.  We added a detailed Motion Dynamics Study and Warm-up Sensitivity Analysis to quantify performance across distinct vs. gradual actions and to analyze the impact of intra-class clustering on curvature methods.  Finally, we revised the caption of Figure 4 and referenced Figure 6 to explicitly demonstrate that boundaries correspond to curvature local minimal (valleys), resolving the visual confusion regarding high-frequency signals.

**II. Experimental Verification, Efficiency, and Evaluation Metrics**
**R-7JNX (W2; Q2, Q3):** Requests for comparisons with adaptive segment division method in the EDD module; inquires about the effect of using curvature only during training; suggests comparing with learning-based boundary detectors  in the CGS module.
**R-t1pj (Major 2; Q3):** Requests an explanation for the phenomenon where the CGS module alone outperforms the Full model on PKU-MMD (X-view).
**R-doPp (W3, W4):** Requests the addition of runtime analysis; questions the justification for using Acc/Edit/F1 instead of IoU as the metric.

**Our summary of responses:**
We conducted a comprehensive efficiency analysis demonstrating that our method achieves faster convergence and higher accuracy with negligible computational overhead.  We also added adaptive segmentation experiments to justify our design choices and clarified the performance trade-offs in X-view settings, explaining how CGS favors continuity while EDD sharpens precision.  Finally, we confirmed that our F1 metrics are inherently IoU-based and validated the CGS module by comparing it against a Gradient-based Saliency baseline, showing the superiority of curvature for STAS.

**III. Symbol definitions, figures/charts/diagrams, and reproduction details**
**R-7JNX (W4):** Asks about the specific feature extraction location (after the Encoder or after the Head) and the curvature calculation method (joint-wise or global).
**R-hoUk (W3):** Believes the description of the foundation model is excessively long.
**R-t1pj (Major 1, 3; Minor 3,4,5; Q1,Q2,Q4):** Points out undefined mathematical symbols or inconsistent dimensions and typo errors; Figure 2 has a confusing flowchart direction and variable occlusion; requests details on curvature calculation in the CGS module when window size w>1.
**R-doPp (W1, W2):** Points out typesetting issues in Figure 1, specifically the font size being too small; points out some typos.

**Our summary of responses:**

We rigorously revised the manuscript to standardize all mathematical notations and corrected typos, while redrawing Figure 2 to improve logical flow and clarity. We streamlined the foundation model description and clarified the windowed curvature calculation, supporting it with a sensitivity analysis for window size w. Finally, we resolved all formatting and typesetting issues to ensure professional presentation.

**IV. Scope & Related Works**
**R-hoUk (W1) & R-t1pj (Minor 2):** The related work section is missing citations.
**R-doPp (Q):** Believes the field of skeletal action segmentation (STAS) is too narrow and suggests broadening the scope of method applications.

**Our summary of responses:**
We updated Section 2 with missing citations and justified the distinct utility of skeletal analysis. Crucially, we validated our method's generalizability by extending it to the RGB-based WTAL and TAS task, demonstrating consistent performance gains beyond skeletal data.

**The specific responses can be found in our detailed point-by-point replies to each reviewer's specific comments.**

---

### Meta-Review · Area_Chair_viqb · 2026-01-08

**Summary:**

The paper proposes CurvSeg, a framework for Skeleton-based Temporal Action Segmentation (STAS). The core problem addressed is the "information silo" created by decoupling action classification and action boundary localization. The authors introduce a geometric curvature guidance mechanism based on the insight that well-learned classification features form compact clusters; trajectories of data sequences exhibit high curvature within an action segment (as they turn to stay within a cluster) and low curvature at transitions (as they move between clusters).

The authors propose curvature-guided synergy (CGS): A bidirectional consistency constraint where curvature signals from classification features guide boundary detection, and localization predictions supervise the classification space to ensure clearer boundaries. The author also introduce expert-driven decoupling (EDD): A Mixture-of-Experts (MoE) framework with dual-expert weighting. It uses classification experts for semantic-oriented representations and localization experts for fine-grained temporal details.

The paper initially received mixed reviews (6, 6, 4, 4). Reviewers recognized the novelty of the geometric insight but also noted that the insight is from a prior art. The experimental evaluation is also acknowledged as showing superior results on multiple benchmarks like PKU-MMD and LARa. Primary concerns included limited failure analysis for smooth transitions, notation inconsistencies and confusing figures and runtime complexity. Reviewer doPp addtionally questions value of the geometric insight in broad domains.

In rebuttal, the authors provided additional evidence and addressed most of the reviewers' concerns. Meta-review feels the paper after revision has improved quality and qualifies for acceptance to ICLR.

**Reviewer Concerns:**

Concerns on curvature visualization, segment partioning, runtime complexity, and writing have been addressed by author rebuttals and revision. The following concerns remain:

- Curvature-based guidance relies on sharp directional changes in the feature trajectory to identify action boundaries. Reviewers noted that "gradual transitions" or smooth, low-dynamic movements (like "Step Sequences") might not produce clear curvature "valleys". While the authors showed that their method improves upon the baseline even in these cases (+6.02% Edit Score), the performance gap between high-dynamic ($72.86\%$) and low-dynamic ($52.86\%$) actions remains significant, representing a persistent challenge for the geometric approach.

- Skeleton sequences obtained from motion capture or pose estimation often contain sensor noise or "jitter". Reviewers were concerned that this could create false curvature peaks or overly smooth "valleys" that confuse the model. The authors introduce a two stage process to suppress noise. However, the authors admitted that the method still faces limitations in handling extremely noisy skeleton data, noting this as a primary area for future works.

- The current evaluation focused on single-person benchmark datasets like PKU-MMD and LARa. The authors mentioned this in their concluding remarks but extending this to complex multi-person scenarios, as noted by reviewers, remains an unaddressed technical challenge.

**Reviewer Scores:**

Reviewer hoUk and 7JNX would likely keep their initial rating.

Reviewer t1pj would likely increase rating as most concerns are addressed in rebuttal.

Reviewer doPp would like keep the initial rating of 4 given the concern on novelty and insight.

---

### Decision · Program_Chairs · 2026-01-26

Accept (Poster)